# SARS-CoV2-mediated suppression of NRF2-signaling reveals potent antiviral and anti-inflammatory activity of 4-octyl-itaconate and dimethyl fumarate

David Olagnier ⬤ et al.[#]

Antiviral strategies to inhibit Severe Acute Respiratory Syndrome Coronavirus 2 (SARS-CoV2) and the pathogenic consequences of COVID-19 are urgently required. Here, we demonstrate that the NRF2 antioxidant gene expression pathway is suppressed in biopsies obtained from COVID-19 patients. Further, we uncover that NRF2 agonists 4-octyl-itaconate (4-OI) and the clinically approved dimethyl fumarate (DMF) induce a cellular antiviral program that potently inhibits replication of SARS-CoV2 across cell lines. The inhibitory effect of 4-OI and DMF extends to the replication of several other pathogenic viruses including Herpes Simplex Virus-1 and-2, Vaccinia virus, and Zika virus through a type I interferon (IFN)-independent mechanism. In addition, 4-OI and DMF limit host inflammatory responses to SARS-CoV2 infection associated with airway COVID-19 pathology. In conclusion, NRF2 agonists 4-OI and DMF induce a distinct IFN-independent antiviral program that is broadly effective in limiting virus replication and in suppressing the pro-inflammatory responses of human pathogenic viruses, including SARS-CoV2.

---

[#]A list of authors and their affiliations appears at the end of the paper.

The 2020 SARS-CoV2 pandemic emphasizes the urgent need to identify cellular factors and pathways that can be targeted by new broad-spectrum antiviral therapies. Viral infections usually cause disease in humans through both direct cytopathogenic effects and excessive inflammatory responses of the infected host. This also seems to be the case with SARS-CoV2, as COVID-19 patients develop cytokine storms that are very likely to contribute to, if not drive, immunopathology, and disease severity[1,2]. For these reasons, antiviral therapies must aim to not only inhibit viral replication but also to limit inflammatory responses of the host.

Nuclear factor (erythroid-derived 2) -like 2 (NRF2) belongs to the cap´n´collar basic leucine zipper family of transcription factors characterized structurally by the presence of NRF2-ECH homology domains[3]. At homeostasis, NRF2 is maintained in an inactive state in the cytosol by association with its inhibitor protein KEAP1 (Kelch-like ECH-associated protein 1), which targets NRF2 for proteasomal degradation[4]. In response to oxidative stress, KEAP1 is inactivated and NRF2 is released to induce NRF2-responsive genes. In general, the genes under the control of NRF2 protect against stress-induced cell death and NRF2 has thus been suggested as the master regulator of tissue damage during infection[5]. Importantly, NRF2 is now demonstrated as an important regulator of the inflammatory response[6,7] and functions as a transcriptional repressor of inflammatory genes in murine macrophages, most notably interleukin (IL-) 1β[8].

Recent reports demonstrated that NRF2 was induced by several cell derived metabolites including itaconate and fumarate, to limit inflammatory responses following lipopolysaccharide stimulation[9]. A chemically synthesized, cell-permeable derivative of itaconate, 4-octyl-itaconate (4-OI) was then shown to be a potent NRF2 inducer[9]. Of special interest is the derivative of fumarate, dimethyl fumarate (DMF), a US Food and Drug Administration approved drug that is used as an anti-inflammatory therapeutic in multiple sclerosis (MS), with the capacity to suppress pathogenic inflammation through a Nrf2-dependent mechanism[10,11].

In addition to limiting the inflammatory response to LPS, 4-OI also inhibited the Stimulator of Interferon Genes (STING) antiviral pathway and interferon (IFN) stimulated gene expression via NRF2 induction[12]. A recent single-cell RNA-sequencing analysis demonstrated that the NRF2 antioxidant gene signature correlated with resistance to HSV1 infection[13]. However, whether NRF2 agonists can be used to inhibit SARS-CoV2 replication or other pathogenic viruses remains unkown.

Here we demonstrate that expression of NRF2-dependent genes is suppressed in biopsies from COVID-19 patients and that treatment of cells with NRF2 agonists 4-OI and DMF induced a strong antiviral program that limits SARS-CoV2 replication. This antiviral program extends to other pathogenic viruses including Herpes Simplex Virus-1 and -2 (HSV-1 and HSV-2), Vaccinia Virus (VACV), and Zika Virus (ZIKV). Furthermore, 4-OI and DMF limits the release of pro-inflammatory cytokines in response to SARS-CoV2 infection and to virus-derived ligands through a mechanism that involves inhibition of IRF3 dimerization. In summary, NRF2 agonists are plausible broad-spectrum antiviral and anti-inflammatory agents. Our results suggest that repurposing the clinically approved compound DMF may represent a rapidly applicable strategy for the treatment of COVID-19-associated disease.

## Results

**The NRF2-response is suppressed in COVID-19 patient biopsies.** To identify host factors or pathways important in the control of SARS-CoV2 infection, publicly available transcriptome data sets including transcriptome analysis of lung biopsies from COVID-19 patients were analyzed using differential expression analysis[14]. Here, genes linked with inflammatory and antiviral pathways, including RIG-I receptor and Toll-like receptor signaling, were enriched in COVID-19 patient samples, whereas genes associated with the NRF2 dependent antioxidant response were suppressed in the same patients (Fig. 1a–c). That NRF2-induced genes are repressed during SARS-CoV2 infections was supported by reanalysis of another data-set building on transcriptome analysis of lung autopsies obtained from five individual COVID-19 patients (Desai et al.[15]) (Fig. 1d). Furthermore, that the NRF2-pathway is repressed during infection with SARS-CoV2 was supported by in vitro experiments where the expression of NRF2-inducible proteins Heme Oxygenase 1 (HO-1) and NAD(P)H quinone oxydoreducatse 1 (NqO1) was repressed in SARS-CoV2 infected Vero hTMPRSS2 cells while the expression of canonical antiviral transcription factors such as STAT1 and IRF3 were unaffected (Supplementary Fig. 1). These data indicate that SARS-CoV2 targets the NRF2 antioxidant pathway and thus suggests that the NRF2 pathway restricts SARS-CoV2 replication.

**NRF2 agonists 4-OI and DMF inhibit SARS-CoV2 replication.** Considering that NRF2 suppresses antiviral IFN-responses, it was surprising to discover that treatment of Vero cells with 4-OI generated before infection with SARS-CoV2 (strain #291.3 FR-4286 isolated from a patient in Germany) resulted in a $10^2$–$10^4$ reduction in SARS-CoV2 RNA levels in a dose dependent manner (Fig. 2a, b) while not affecting cell viability, as determined by lactate dehydrogenase (LDH) release assay (Supplementary Fig. 2). Furthermore, subsequent release of progeny SARS-CoV2 virus particles to the cell supernatant was equally decreased by 4-OI treatment. This was measured by TCID50 assay to quantify virus by dilution of virus-induced cytopathogenic effects and by plaque assay (Fig. 2c–f). The reduced viral replication led to reduced virus-induced cytotoxicity of the infected Vero cells as determined by LDH release assay and immunoblotting for cleaved Caspase 3 and Poly(ADP-Ribose) Polymerase 1 (PARP-1), which are hallmark indicators of apoptosis[16] (Fig. 2g, h). Interestingly, the observation that the NRF2 pathway is inhibited in response to SARS-CoV-2 infection was recapitulated in SARS-CoV-2 infected Vero cells, as demonstrated by both the decrease in the basal level expression of NRF2-driven proteins HO-1 and NqO1 and their inability to be induced by the Nrf2 agonist 4-OI (Fig. 2h). The antiviral effect of 4-OI was also demonstrated in the lung cancer cell line Calu-3, where SARS-CoV2 RNA levels were reduced by >2-logs (Fig. 2i), while release of progeny virus was reduced by >6-logs based on TCID50 analysis of cell supernatants (Fig. 2j, k). In the immortalized human epithelial cell line NuLi, total infection levels were relatively low compared to what was observed in Calu3 and Vero cells, but 4-OI treatment still reduced SARS-CoV2 RNA levels and release of progeny virus (Fig. 2l, m). We further tested the antiviral effect towards SARS-CoV2 in primary human airway epithelial (HAE) cultures (Fig. 2n). Here, 4-OI treatment also significantly reduced viral RNA levels (Fig. 2o). Interestingly, DMF treatment of Calu3 cells likewise inhibited SARS-CoV-2 replication by a similar magnitude as observed with 4-OI (Fig. 2p-q); in Vero cells, a reduced but significant inhibition of SARS-CoV-2 multiplication was also detected (Fig. 2r). To further evaluate the role of the NRF2/KEAP1 axis in SARS-CoV-2 replication, siRNA silencing of KEAP1 was used to activate NRF2. Silencing of KEAP1 decreased SARS-CoV-2 RNA levels, reduced viral protein expression, and virus titers (Fig. 2s–u). Thus, both treatment with NRF2 agonists and genetic activation of NRF2 led to restriction of SARS-CoV2 replication. Finally, the antiviral effect of 4-OI was reproduced using a different SARS-CoV-2 isolate from Japan[17] (Fig. 2v, w).

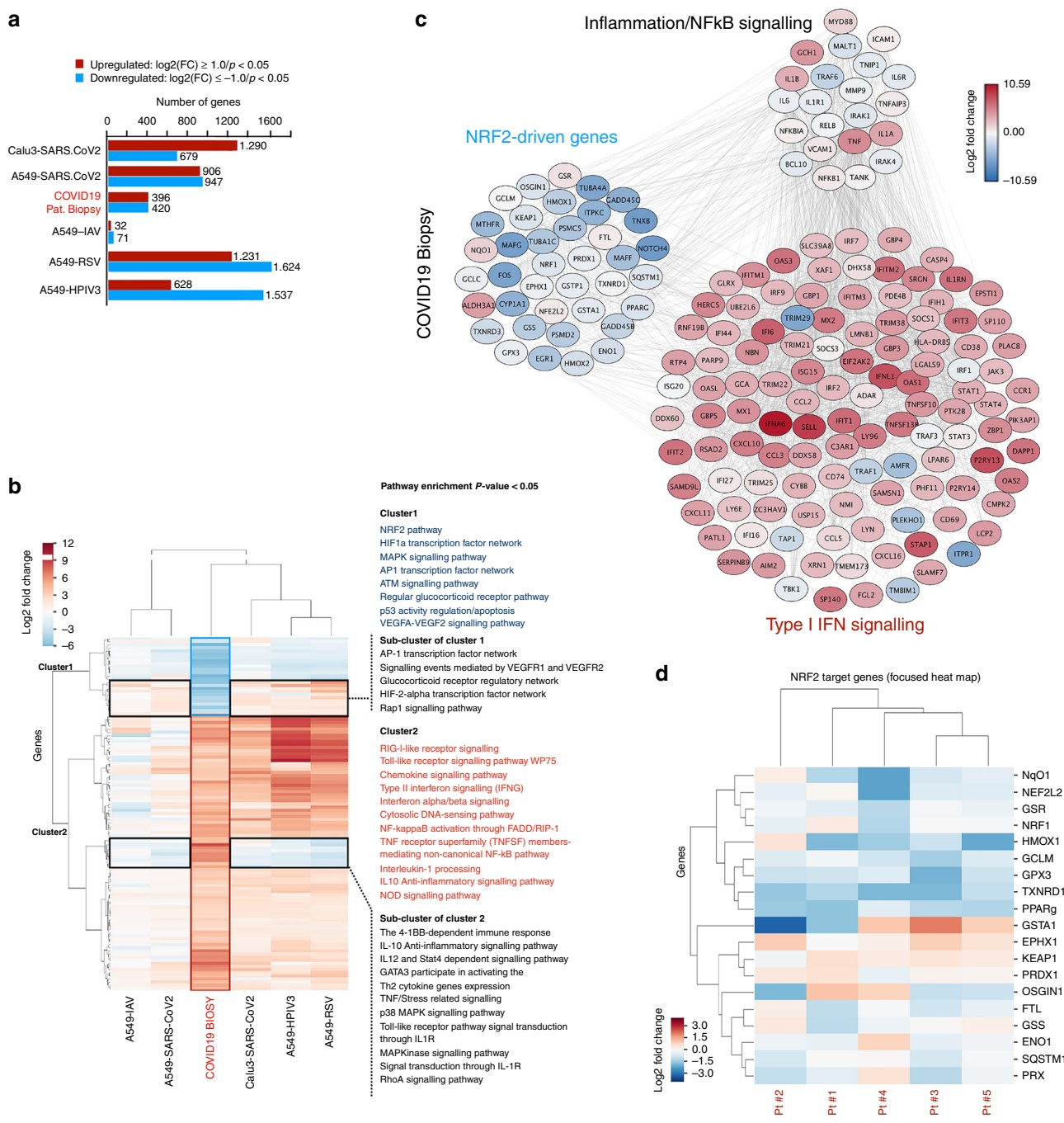

**Fig. 1 Expression of NRF2-driven genes is suppressed in COVID-19 patient biopsies. a, b, c** Reanalysis of data published by Blanco-Melo et al.[14] (**a**) Bar-chart of the number of transcripts showing significant differential expression (adjusted $p$ value < 0.05 and log2(FC) > 1.0). Data from COVID19 lung biopsies was normalized against healthy lung biopsies, and in cell lines Calu3, NHBE, and A549 infected with either SARS-CoV2, Influenza A virus (IAV), Respiratory Syncytial virus (RSV), or human parainfluenza virus type 3 (HPIV3) against mock treated cells. Expression and $p$ values were calculated with DESeq2 using Wald test statistic and Benjamini-Hochberg correction for multiple testing. **b** Heat map of the subset of genes significantly differentially expressed in COVID19 biopsies and simultaneously differentially expressed in at least 3 of the other conditions tested. The genes in each cluster were used for pathway enrichment analysis. Genes in cluster 1 are dominantly down-regulated in COVID19 biopsies while a sub-cluster of genes in cluster 1 are up-regulated in the cell lines. Conversely, genes in cluster 2 are predominantly up-regulated in biopsies and in most other test-samples. A subcluster of the genes in cluster 2 are down regulated in the cell lines. For each cluster, the significantly enriched pathways are listed (EnrichR). **c** Cloud analysis of NRF2-driven differentially expressed genes. Subsets annotated as inflammation/NFκB signaling and Type I IFN signaling exhibit different expression patterns. The experiment is a reanalysis of data from Blanco-Melo et al. [https://doi.org/10.1101/2020.03.24.004655]. **d** Reanalysis of the data from Desai et al.[15] GEO accession code GSE150316. Heat map of NRF2 target genes of the lung autopsies from the five COVID19 patients. Healthy lung samples were used as negative control.

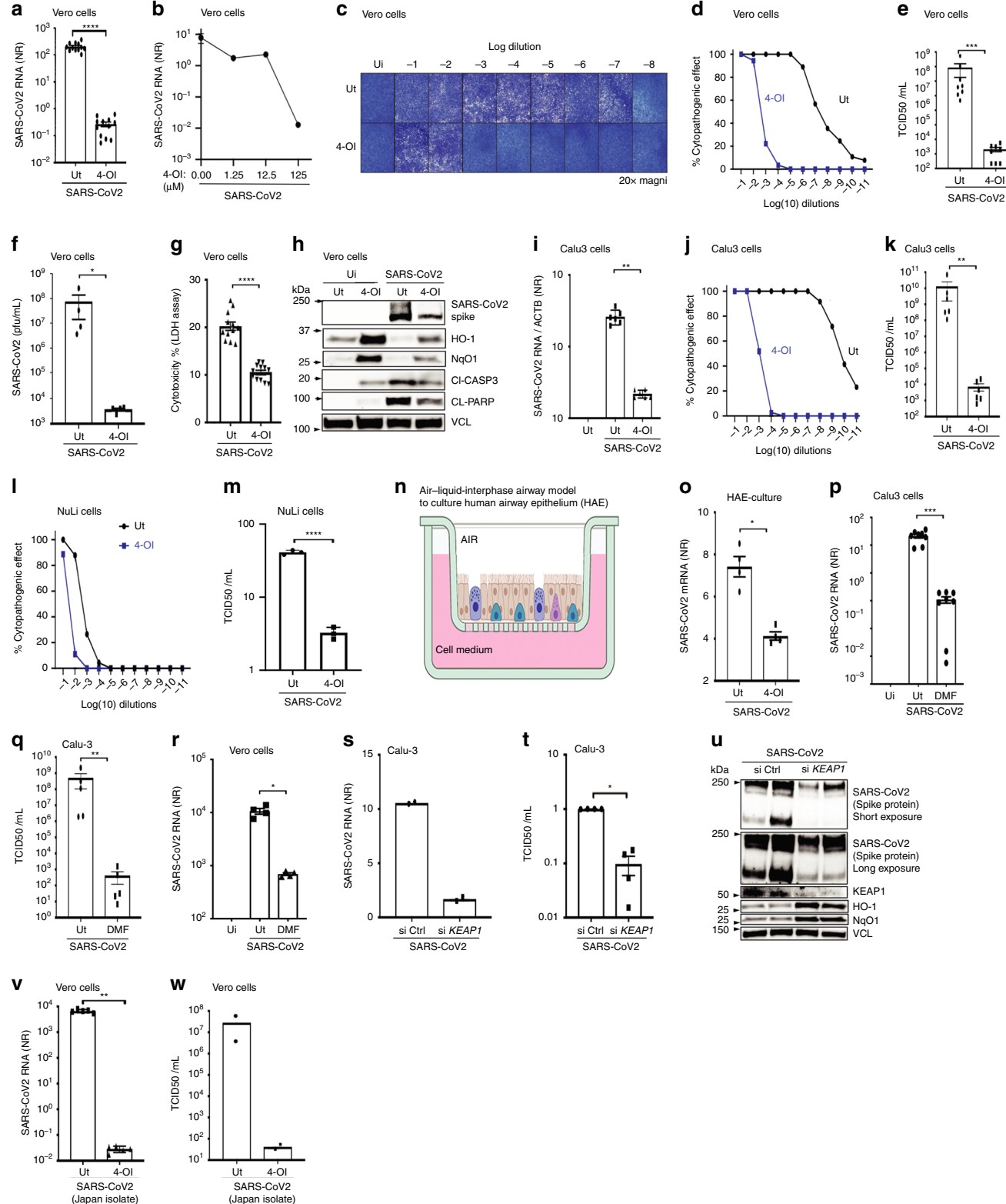

These data demonstrate that NRF2 inducers 4-OI and DMF inhibit SARS-CoV-2 replication by a magnitude of several logs.

**4-OI broadly inhibits viral replication through an IFN-independent pathway.** The antiviral effect of 4-OI was not restricted to SARS-CoV2 but also extended to other human pathogenic viruses. Using the human keratinocyte cell line HaCaT, 4-OI also inhibited HSV1 and HSV2 multiplication, as measured by virus titers, intracellular viral RNA content and accumulation of viral proteins (Fig. 3a–e and Supplementary Fig. 3). In contrast, the NRF2-inducible genes HO-1, NqO1, and Sequestosome 1 (SQSTM1) were highly increased in response to 4-OI treatment (Fig. 3c and Supplementary Fig. 3); the antiviral effect of 4-OI was at least partly dependent on NRF2, since siRNA silencing of NRF2 reduced the capacity of 4-OI to inhibit HSV1 infection and viral protein expression (Fig. 3f-g). To determine

**Fig. 2 4-Octyl-itaconate (4-OI) and dimethyl fumarate (DMF) inhibit SARS-CoV2 replication.** (**a**–**f**) **h** TMPRSS2-Vero cells treated with 4-OI (48 h) and infected with SARS-CoV-2 (48 h). Viral replication by qPCR (**a**, **b**), TCID50-assay (**c**–**e**) or plaque assay (**f**). Are pooled data from four (**a**) and three (**e**) independent experiments in duplicates and triplicates. Data-points represent one biological sample. Data in (**b**), (**c**), and (**d**) are representative of two independent experiments. (**f**), supernatants from (**e**) analysed by plaque assay. Data are pooled from two independent experiments in duplicates. Data-points represent one biological sample. **g** TMPRSS2 cells with 4-OI and infected with SARS-CoV-2 for 48 h before LDH release-assay. Data pooled from two independent experiments in sextuplicates. **h** cells from (**g**) immunoblotted. Blot representative of two independent experiments. Calu3 cells with 4-OI and infected with SARS-CoV-2. Replication by qPCR (**i**) and TCID50 (**j** + **k**). Data pooled from two independent experiment in triplicates. **l**–**m**, NuLi cells treated with 4-OI and infected with SARS-CoV-2 for 48 h. Viral replication by TCID50. Data representative of two independent experiments in duplicates. **n**, HAE-culture graphic. **o**, HAE-cultures were treated with 4-OI overnight and infected with SARS-CoV-2. Viral replication by qPCR. **p**–**r**, Calu-3 and Vero hTMPRSS2 cells treated with DMF (150-200 μM) and infected with SARS-CoV-2. **p** is pooled data from three independent experiments in duplicates and triplicates. **q** is pooled data from two independent experiments in duplicates and triplicates. **r** is pooled data from two independent experiments in duplicates. siRNA treated Calu-3 cells infected with SARS-CoV2. Viral replication by qPCR (**s**), TCID50 (**t**), or immunoblotting (**u**). **s** is one representative of two independent experiments. **t** is pooled data from two independent experiments in duplicates. **v**, **w**, Vero cells treated with 4-OI at 125 μM and infected with SARS-CoV2 from Japan. Data in (**u**) is representative of two independent experiments, and (**v**) is pooled data from two independent experiments in duplicates and triplicates. Data in (**w**) is data from one experiment with two biological samples. Bars indicate mean ± s.e.m. Unless otherwise stated, statistical analyses by two-tailed Mann–Whitney (*$p < 0.05$, **$p < 0.01$, ***$p < 0.001$, and ****$p < 0.0001$). P values: (a: $p < 0.0001$, e: $p = 0.0002$, f: 0.0286, g: $p < 0.0001$, i: $p = 0.0022$, k: $p = 0.0022$, m: $p < 0.0001$(two-tailed t test), o- $p = 0.0268$, p: $p = 0.0002$, q: $p = 0.0079$, r: $p = 0.0286$, t: $p = 0.0286$, v- $p = 0.0079$. Source data are provided as a Source Data file.

the range of pathogenic viruses inhibited by 4-OI, HaCaT cells and bone marrow derived dendritic cells (BMDCs) were pre-treated with 4-OI and then infected with vaccinia virus (VACV) or electromelia virus (ECTV); infection and replication of both poxviruses were strongly inhibited by 4-OI (Fig. 3h-n). Further the inhibitory effect of 4-OI extended to zika virus (ZIKV), an important human pathogenic virus causing a severe disease when transmitted in utero[18], where 4-OI pre-treatment reduced zika virus replication in both A549 and Huh-7 cells (Fig. 3o-p). Interestingly, the antiviral effect of 4-OI did not extend to infection with vesicular stomatitis virus (VSV) (Supplementary Fig. 4); this result further provided indirect evidence that the antiviral program induced by 4-OI was distinct from the classical IFN-dependent antiviral response, since VSV, and especially VSVd51M used in this study, is highly sensitive to the effects of IFNs. The antiviral effect of 4-OI relied on intracellular restriction of replication, since HSV entry was not inhibited by 4-OI treatment across cell lines (Supplementary Fig. 5).

To address the possibility that the inhibitory effect of 4-OI was independent of type I IFN signaling, as suggested by the results in Vero cells that are deficient in type I IFN[19], HaCaT cells deficient in IFN alpha receptor 2 (IFNAR2), Signal Transducer and Activator of Transcription 1 (STAT1) or STING were treated with 4-OI, followed by infection with HSV1 and VACV.

Replication of both viruses was inhibited by 4-OI in IFNAR2 and STAT1 KO cells; for HSV1 replication was also decreased in STING KO cells, as measured by plaque assay and expression of viral proteins (Supplementary Fig. 6). In conclusion, 4-OI induced an antiviral program that functioned independently of IFN signaling (Supplementary Fig. 6).

To examine what general pathways were affected by 4-OI treatment that could predict the underlying mechanism of its antiviral effect, RNA sequencing analysis was performed with HaCaT cells treated with 4-OI and then infected with HSV1. By comparing untreated to 4-OI treated cells, either with or without infection with HSV1, several pathways were identified that were induced or repressed by 4-OI, including repression of the IFN-signaling pathway by 4-OI (Supplementary Fig. 7). Amongst the top up-regulated genes induced by 4-OI was the heme oxygensase-1 (HO-1, *HMOX-1*), an enzyme canonically involved in stress detoxification, also reported to have antiviral activity against amongst other viruses Zika and Dengue viruses[20–22]. To assess whether, HO-1 had any antiviral activity in our cellular system, Vero hTMPRSS2 and Calu-3 cells were either transfected with an overexpression plasmid encoding HO-1, or genetically

silenced for KEAP1 and HO-1 by siRNA, respectively before infection with SARS-CoV-2. These treatments (HO-1 over-expression or silencing) did not alter SARS-CoV-2 infection/replication, suggesting an HO-1-independent antiviral program induced by 4-OI (Supplementary Fig. 8).

In an attempt to pin-point the antiviral mode of action of 4-OI, we also used microscope-based analysis of morphology by cell-paint technology (Supplementary Fig. 9) to compare morphological changes in cells treated with 4-OI to cells treated with compounds that have known cellular targets and with cells treated with other compounds with reported antiviral activity toward SARS-CoV2 including Remdesivir and Hydroxychloroquine[23,24]. In this analysis, 4-OI had a low but significant morphological activity whithout loss of cell viability. Interestingly, the activity of 4-OI did not seem to overlap with other compounds known to perturb cell morphology, including Rapamycin, Bafilomycin, Tunicamycin, Cyclohexamide, Emetine, Mitomycin, or Doxorubicin. Interestingly, there was also no observable overlap with the activity profile of Remdesivir or hydroxychloroquine, indicating that the antiviral mode of action of 4-OI is distinct from known antiviral mechanisms.

**4-OI and DMF suppress the inflammatory response to SARS-CoV2.** In COVID-19, an uncontrolled proinflammatory cytokine storm contributes to disease pathogenesis and lung damage[25]. For this reason, we investigated if 4-OI and DMF inhibited inflammatory cytokine gene expression induced by SARS-CoV2. In Calu-3 cells, SARS-CoV2 infection increased the expression of *IFNB1*, C-X-C motif chemokine 10 (*CXCL10*), Tumor Necrosis Factor alpha (*TNFA*), *IL-1B*, and C-C chemokine ligand 5 (*CCL5*). Interestingly, this induction was abolished by 4-OI pre-treatment, thus reducing the pro-inflammatory response to SARS-CoV2 (Fig. 4a-b). In contrast, expression of the NRF2 inducible gene *HMOX1* was highly increased in response to 4-OI treatment (Fig. 4c). The potential anti-inflammatory effect of 4-OI in this context was supported when using HAE cultures. Here, treatment with 4-OI also reduced the expression of *IFNB1*, *CXCL10*, *TNFA*, and *CCL5* in the context of SARS-CoV2 infection (Fig. 4d-e), while increasing the expression of the NRF2 inducible gene *HMOX1* (Fig. 4f). A similar pattern was observed in Calu3 cells, treated with DMF before SARS-CoV2 infection; *IFNB1*, *CXCL10*, and *CCL5* mRNA levels were reduced in DMF treated cells, while *TNFA* mRNA levels were unaffected (Fig. 4g, h). In contrast, DMF treatment increased the expression of the NRF2 inducible gene *HMOX1* (Fig. 4i).

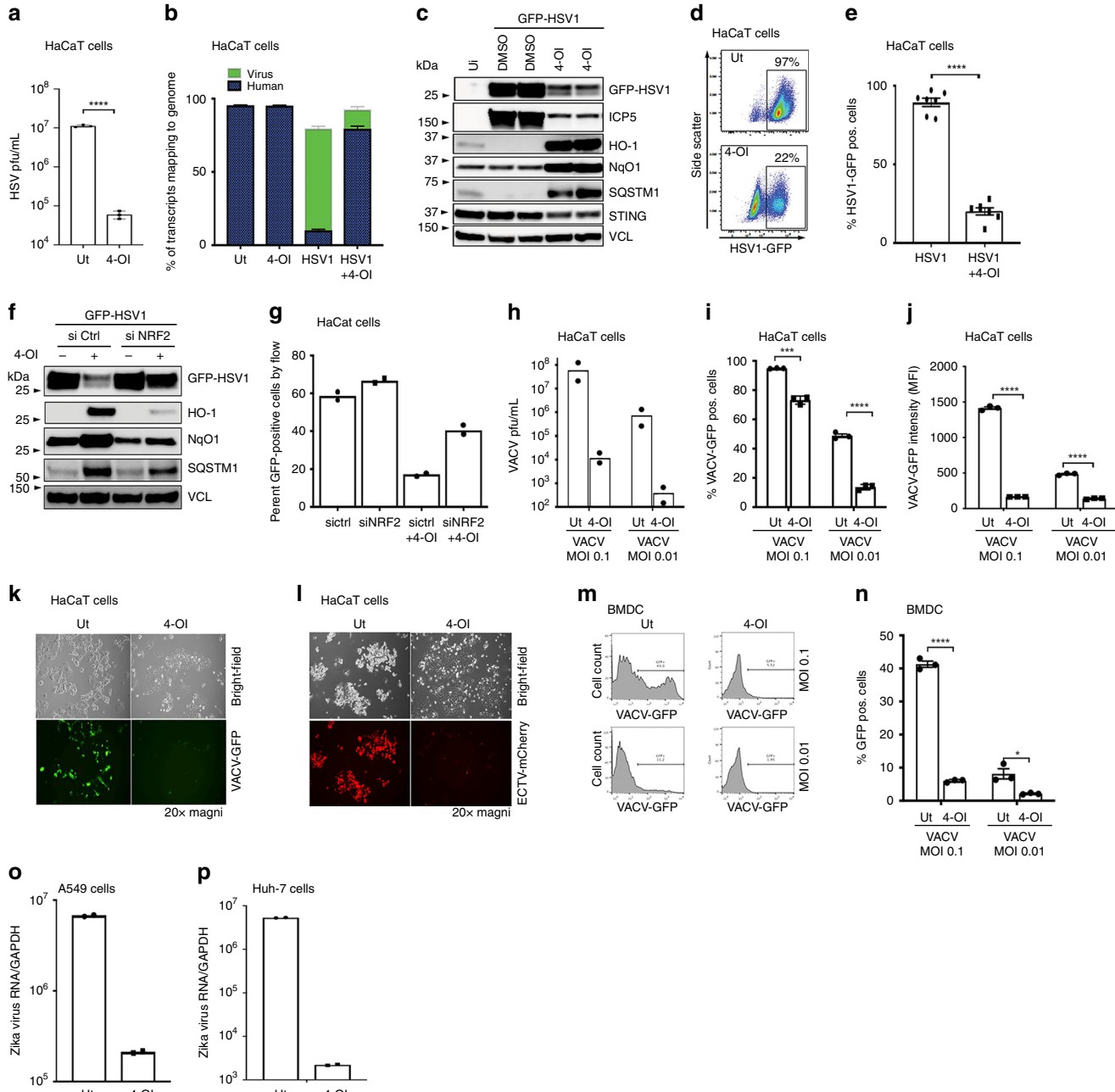

**Fig. 3 4-OI broadly inhibits other pathogenic viruses including HSV, VACV, and Zika Virus. a** HaCaT cells treated with 4-OI (125 μM) for 48 h and infected with HSV1-GFP (MOI 0.01). Viral titers were determined by plaque assay. Data are obtained from one experiment of at least seven independent experiments each performed in triplicates. **b** RNA analyzed using RNA-sequencing ($n = 3$) performed once. **c–e** HaCaT cells treated with 4-OI (125μM) and infected with HSV1-GFP (MOI 0.01). Lysates were analyzed by immunoblotting with Vinculin (VCL) as loading control and by flow cytometry ($n = 7$ from three independent experiments). **f, g** HaCaT cells were lipofected with siRNA for 72 h, subsequently challenged with 4-OI (125 μM) before HSV1-GFP infection (MOI 0.01) for 24 h. Infectivity and silencing efficiency was determined by immunoblotting (**f**) and flow-cytometry (**g**). Data obtained from one experiment representative of three independent experiments. (**h–n**) HaCaT cells (**h-l**) and BMDCs (**m, n**) were treated with 4-OI (125 μM) for 48 h and infected with VACV expressing either GFP or ECTV expressing mCherry for 24 h. Viral titers and infectivity were determined by plaque assay (**h**), flow cytometry (**i-j** and **m-n**), and visualized by confocal microscopy (**k-l**). Data in (**i, j**), and (**n**) are obtained from one experiment representative of two independent experiments each performed in triplicates. **o-p** A549 and Huh-7 cells were pre-treated with 4-OI for 48 h (150 μM) and infected with Zika virus (ZIKV) (MOI 0.1) for 4 days. Viral genome was determined by qPCR. Data were obtained from one experiment representative of two independent experiments. For all panels, bars indicate mean ± s.e.m. Unless otherwise stated, all statistical analysis were performed using a two-tailed Mann–Whitney test to determine statistical significance where *$p < 0.05$, **$p < 0.01$, ***$p < 0.001$, and ****$p < 0.0001$. Individual p values were, **a**: $p < 0.0001$, **e**: $p < 0.0001$, **i**: $p = 0.0002$(two tailed $t$ test, MOI 0.1) and $p < 0.0001$(two tailed $t$ test, MOI 0.01), **j**: $p < 0.0001$(two tailed $t$ test, MOI 0.1) and $p < 0.0001$ (two tailed $t$ test, MOI 0.01), **n**: $p < 0.001$(two tailed $t$ test, MOI 0.1) and $p = 0.0178$(two tailed $t$ test, MOI 0.01). Source data are provided as a Source Data file.

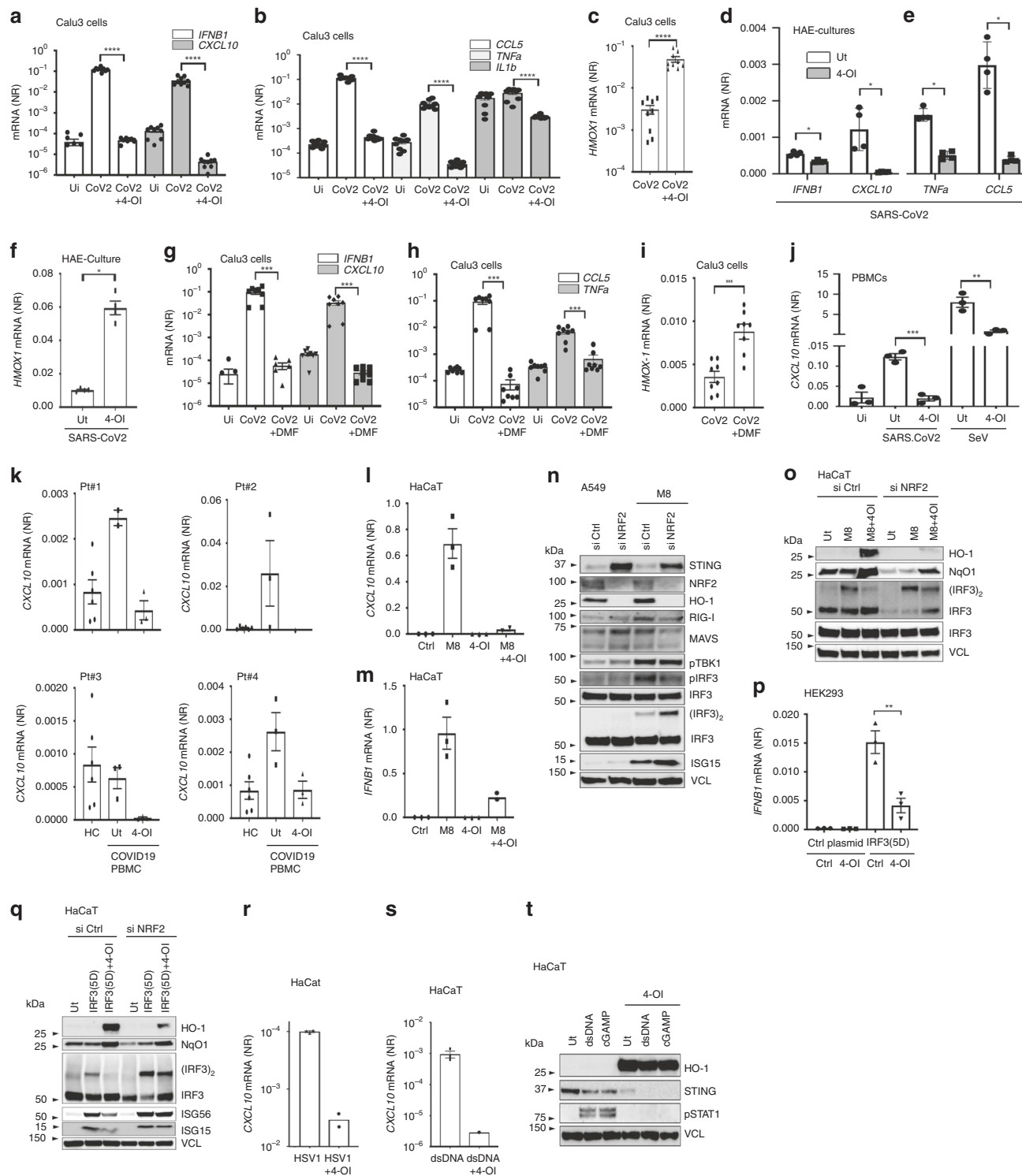

The effect of 4-OI on peripheral blood mononuclear cells (PBMC) harvested from healthy donors was also examined; although PBMC infection by SARS-CoV2 (MOI 1) yielded a very weak induction of *CXCL10* compared to sendai virus (SeV) infection (50 HAU), and no detectable induction of other cytokines, 4-OI treatment also reduced *CXCL10* mRNA levels in this context (Fig. 4j). Furthermore, using PBMCs from four individual patients with severe COVID-19 and admitted to hospital Intensive Care Units, we demonstrated that in three of four patients, the expression of *CXCL10* was increased compared to healthy controls; in all four patients, these levels were reduced

to basal levels after treatment with 4-OI (Fig. 4k), indicating that 4-OI was able to reverse the inflammatory response in patients PBMCs ex vivo. That CXCL10 levels is a relevant readout in SARS-CoV2 was recently supported by a report from Cheemarla et al.[26] ([https://doi.org/10.1101/2020.06.04.20109306])[26] demonstrating that CXCL10 expression is increased in the upper airways of patients infected with SARS-CoV2.

The observed decrease in antiviral and proinflammatory cytokine response can be explained by the 4-OI mediated reduction in intracellular viral RNA, with subsequent reduced induction of cytokines through cellular RNA sensors such as

**Fig. 4 4-OI and DMF limit SARS-CoV2 and HSV induced inflammatory responses. a–c** Calu-3 treated with 4-OI (125 μM, 48 h) before SARS-CoV2 infection (MOI 0.5, 48 h) and qPCR analysis. a, b, and c are pooled data from three independent experiments in triplicates. **d-f** HAE-cultures treated with 4-OI before SARS-CoV2 infection for 24 h and analysis by qPCR. Data are from two donors. **g-i** Calu-3 cells with DMF (48 h) before SARS-CoV2 infection. Analysis by qPCR. In (**g, h,** and **i**), display pooled data from three independent experiments in duplicates and triplicates. (**d-f**), data are representative of four independent HAE-cultures. (**j**) Healthy PBMCs treated with 4-OI before infection and qPCR analysis. Data are representative of two donors. (**k**) PBMCs from four COVID-19 patients (**k**) and 2 healthy controls (HC) treated with 4-OI before qPCR. (**l-m**) HaCaTs treated with 4-OI before stimulation with M8 for 6 h followed by qPCR. Data represent one experiment in triplicate. **n** HaCaT cells with siRNAs before M8 treatment for 3 h followed by immunoblotting. Data are representative of two independent experiments. **o** HaCaTs treated with siRNA for 72 h before 4-OI and stimulation with M8. Analysis by immunoblotting. Data representative of two independent experiments. **o-q** HEK293 and HaCaTs transfected with expression-plasmids before 4-OI. In (**q**), HaCaTs were treated with siRNAs for 72 h before plasmid transfection. Analysis by qPCR and immunoblotting. Data in (**p**) are from one experiment performed in triplicate. Data displayed in (**q**) is representative of two independent experiments. (**r-t**) HaCaTs treated with 4-OI before infection with HSV1 or transfection with dsDNA. Analysis by qPCR and immunoblotting at 6 and 3 h respectively. (**n-t**), data are from one experiment representative of two independent experiments. Data in (**r**) and (**s**) are representative of two independent experiments in triplicates. Bars indicate mean ± s.e.m. Unless otherwise stated, all statistical analysis by two-tailed Mann–Whitney test where *$p < 0.05$, **$p < 0.01$, ***$p < 0.001$, and ****$p < 0.0001$. $P$ values:, a: $p < 0.0001$(IFNB1) and $p < 0.0001$(CXCL10), b: $p < 0.0001$(CCL5) and $p < 0.0001$(TNFA) and $p < 0.0001$(IL1B), c: $p < 0.0001$, d: $p = 0.0286$ (IFNB1) and $p = 0.0286$(CXCL10), e: $p = 0.0286$(TNFA) and $p = 0.0286$(CCL5), f: $p = 0.0286$, g: $p = 0.0003$(IFNB1) and $p = 0.0002$(CXCL10), h: $p = 0.0002$(CCL5) and $p = 0.0003$(TNFA), i: $p = 0.0011$, j: $p = 0.0005$ (two-tailed $t$ test, SARS-CoV2) and $p = 0.0053$ (two-tailed $t$ test, SeV), p: $p = 0.0092$ (two-tailed $t$ test). Source data are provided as a Source Data file.

RIG-I. We therefore investigated the effect of 4-OI on the induction of IFN and of IFN stimulated genes responses activated by a sequence optimized RIG-I agonist, previously reported as M8[27]. Interestingly, 4-OI treatment reduced the IFN-responses induced by RIG-I stimulation (Fig. 4l-m), by inhibiting Interferon Regulatory Factor 3 (IRF3) dimerization but did not inhibit the upstream phosphorylation of Tank Binding Kinase 1 (TBK1) or IRF3 expression itself (Fig. 4n). Importantly, NRF2 expression itself was associated with the inhibition of IRF3 dimerization and host antiviral gene expression, since NRF2 silencing was sufficient to restore IRF3 dimerization and limit the inhibitory effect of 4-OI (Fig. 4n-o). Furthermore, using the constitutively active form of IRF3, IRF3(5D)[28], 4-OI still blocked IRF3 dimerization—an effect eliminated by NRF2 silencing (Fig. 4p-q). These data indicate that an NRF2 inducible mechanism interferes with IFN induction by inhibiting IRF3 dimerization and activation. We have previously reported that 4-OI inhibited the expression of STING, which is important for the induction of the IFN-response in cells stimulated with cytosolic DNA[12]. In line with these previous results, 4-OI inhibited the IFN antiviral response to HSV1 infection and to stimulation with STING agonists dsDNA and cGAMP (Fig. 4r–t).

## Discussion

Altogether, this study demonstrates that the expression of NRF2 dependent antioxidant genes is significantly inhibited in COVID-19 patients, and that the NRF2 agonists 4-OI and DMF inhibit both SARS-CoV2 replication, as well as the expression of associated inflammatory genes. Significantly, DMF is currently used as an anti-inflammatory drug in relapsing-remitting MS and could easily be repurposed and tested in clinical trials as a small molecule inhibitor of SARS-CoV-2 replication and inflammation-induced pathology in COVID19 patients.

The ability of NRF2 inducers to also reduce potentially pathogenic IFN- and inflammatory responses while retaining their antiviral properties is unique to these compounds and highlights their potential to control virus-induced pathology. That NRF2 might be a natural regulator of IFN-responses in the airway epithelium is supported by a recent report demonstrating that NRF2 activity is high while IFN activity is low in the bronchial epithelium[29]. Our discovery that NRF2 seems to affect stability of IRF3 dimerization adds to the notion that NRF2 is a more universal regulator of type I IFN induction, while keeping in mind that this is not the only mechanism through which this could be achieved. Other examples are the suppression of type I

IFNs by NRF2 through inhibition of STING expression[12] and the regulation of IFN-responses to VSV through NRF2-mediated autophagy[30]. That VSV is highly sensitive to IFNs and its outgrowth promoted by autophagy could be part of the explanation as to why VSV, in contrast to other viruses tested, was insensitive to or even seemed to benefit from a chemical pre-treatment with 4-OI.

The observation that 4-OI strongly inhibited the IFN antiviral response to both cytosolic DNA and RNA but retained the capacity to block viral replication also supports the existence of an unidentified cellular program that restricts viral replication independently of IFNs. This is supported by the negative correlation between NRF2-inducible gene expression and infection with HSV1 discovered by Wyler et al., through single cell transcriptome analysis[13]. By analogy with the induction of hundreds of IFN-stimulated effector genes, an NRF2 antiviral program may also use a variety of mechanisms to restrict viral replication by targeting distinct stages of viral replication. Our transcriptome analysis of the cells treated with 4-OI identified several cellular pathways or programs that each could contribute to the antiviral state of the 4-OI treated cells. These cellular pathways included Hypoxia-inducible-factor 1 alpha-controlled gene expression program as well as basal cellular glycerolipid and nitrogen metabolism. Further studies will be needed to unravel the contribution of each of these cellular pathways to antiviral immunity.

Considering the dual effect of NRF2 agonists on both viral replication and inflammatory markers, it would be interesting to investigate if patients that develop severe SARS-CoV2 pathology also have an underlying dysregulation of NRF2 or of central components in cellular pathways that are induced by 4-OI and DMF. This could perhaps be a contributing factor to the reduced control of viral replication and excess inflammatory responses experienced by these patients. Furthermore, it could be valuable to investigate if patients already in DMF therapy have altered susceptibility to SARS-CoV2 infection and if those infected have milder symptoms/and or limited cytokine storm. Finally, the fact that 4-OI and DMF effectively suppressed replication of several human pathogenic viruses illustrates the possibility that repurposing metabolic-derived compounds should be evaluated as broad-spectrum antiviral agents for protection against a range of seasonal and pandemic viral infections.

## Methods

**Cell lines, reagents and culture conditions.** Human lung adenocarcinoma epithelial A549 cells, immortalized human HaCaT keratinocytes, Calu-3 epithelial

lung cancer cells, and human embryonic kidney HEK293T cells were kindly provided by S.R.P. (Aarhus University, Denmark) and cultured in DMEM (Lonza) supplemented with 10% heat inactivated fetal calf serum, 200 IU.mL$^{-1}$ penicillin, 100 µg.mL$^{-1}$ streptomycin and 600 µg.mL$^{-1}$ glutamine (hereafter termed DMEM complete). Vero E6 cells expressing hTMPRSS2 were a kind gift of Makoto Takeda (University of Tokyo, Japan)[17] and were cultured in DMEM (Lonza) supplemented with 10% heat inactivated fetal calf serum, 200 IU.mL$^{-1}$ penicillin, 100 µg.mL$^{-1}$ streptomycin, 600 µg.mL$^{-1}$ glutamine, and 10 µg.mL$^{-1}$ blasticidin. All cell lines were regularly tested for mycoplasma contamination by sequencing from GATC Biotech (Germany). 4-octyl-itaconate (4-OI) was chemically synthetized by Thomas B. Poulsen (Aarhus University, Denmark)[12] and was dissolved in DMSO. DMF was obtained from Sigma (cat# 242926).

To obtain BMDCs, a cell suspension from femurs of C57BL/6 mice (Charles River) was cultured for 6-8 days at 37 °C in RPMI-1640 medium supplemented with 3% FBS and 10% of J558 cell line supernatant containing GM-CSF. Cells were seeded at a density of $10^6$ cells ml-1 and medium was partially replaced every 2 days. BSC-1 cells (ECACC) were maintained in DMEM supplemented with 5% FBS, 2 mM glutamine, 200 IU ml$^{-1}$ penicillin, and 100 IU ml$^{-1}$ streptomycin at 37 °C 5% CO2.

For generation of KO cell line clones in HaCaT cells specific guide RNA sequences targeting STING (5'-AGAGCACACTCTCCGGTACC-3') or STAT1 (5'-TTAATGATGAACTAGTGGAG-3') were cloned into the plasmids pX461 (Addgene) (STING) or LentiCRISPR v2 (Addgene) (STAT1). Wildtype HaCaT cells were transfected with the plasmids using the Lipofectamine 2000 Reagent (Invitrogen, Life Tecnologies). 72 h post transfection, the GFP expressing cells were sorted as single cells by FACS and clones were grown to larger cultures (STING). Or 24 h post transfection, the cells were seeded in a dilution sufficient to obtain single cells clones after the puromycin selection. The 2 µg mL$^{-1}$ puromycin selection was initiated 48 h post transfection and continued for 72 h (STAT1). Hereafter, single cell clones were grown to larger cultures which were validated for absence of protein by western blotting and functional analysis to confirm the biological effect of the gene deficiency.

**Viruses.** We used the SARS-CoV2 strain #291.3 FR-4286 isolated from a patient in Germany, and kindly donated by professor Georg Kochs (Freiburg). The virus was propagated in Vero cells expressing human TMPRSS2[31]. Validation and SARS-CoV2 genome detection was performed with Taqman based qPCR using SARS-CoV2 specific primers and probes with the following sequences: Forward primer: AAATTTTGGGGACCAGGAAC, reverse primer: TGGCACCTGTGTAGGTC AAC, Probe: FAM-ATGTCGCGCATTGGCATGGA-BHQ. HSV-1 KOS strain expressing GFP (HSV-1–GFP), HSV-2 333 strain and HSV-2 MS strain were kindly provided by Søren R. Paludan (Aarhus University, Aarhus, Denmark). All HSVs were propagated in Vero cells, purified by ultra-centrifugation, and titrated by standard plaque assay[32]. HaCat cells were infected with the different HSVs at a multiplicity of infection (MOI) of 0.01 in a small volume of serum-free medium for 1 h at 37 °C. Prior to analysis, cells were incubated with complete DMEM for an additional day of culture. VACV Western reserve strain (VACV-WR) was a recombinant vaccinia virus (VACV) named vtag2GFP expressing the tag2GFP under the control of strong synthetic VACV early/late promoter was kindly provided by Dr. Rafael Blasco (INIA, Spain). VACV-WR and vtag2GFP stocks were semi-purified by centrifugation through a 36% sucrose cushion and titrated twice by plaque assay. The Brazilian ZIKV isolate ZIKV/H.sapiens/Brazil/PE243/2015 was originally described in[33] and was grown on Vero cells. Viral titers were determined by plaque assay on A549 BVDV NPro cells (kind gift from R. Randall, St Andrews). These cells are optimized for virus growth as they stably express the NPro protein of bovine viral diarrhea virus (BVDV), which induces degradation of IRF3[34].

**Viral entry assay.** Quantification of HSV-1 entry in the presence of 4-OI was performed using the cold binding assay[35]. Cells were pretreated with 4-OI (150 µM) or DMSO (control) for 48 h. Cells were pre-incubated at 4 °C for 30 min, then incubated with HSV-1 at a MOI of 10 for 1 h at 4 °C. Cells were then shifted to 37 °C for 1 h to activate virus internalization. After that, cells were washed twice with PBS, then uninternalized virus particles were washed with citric acid buffer (135 mM NaCl, 10 mM KCl, 40 mM citric acid, pH 3) incubation for 5 min, then cells were washed twice more with PBS. Cells were scraped and genomic DNA was extracted using QIAamp DNA mini kit (QIAGEN). Quantitative PCR were performed using UL30-F and UL30-R primers for HSV-1 genomic DNA. Primer sequence: UL30-F: ACATCATCAACTTCGACTGG, UL30-R: CTCAGGTCCT TCTTCTTGTCC.

**Primary cells and culture conditions.** Peripheral Blood Mononuclear cells (PBMCs) were isolated from healthy donors (blood donors gave written consent as accordingly to the ethical guidelines at Aarhus University Hospital) by Ficoll Paque gradient centrifugation (GE Healthcare). Monocytes were separated using a monocyte enrichment kit (STEMCELL) according to the manufacturer's instructions or from PBMCs by adherence to plastic in RPMI 1640 supplemented with 10% AB-positive human serum. Differentiation of monocytes to macrophages was achieved by culturing in Dulbecco's Modified Eagle Medium (DMEM)

supplemented with 10% heat inactivated AB-positive human serum 9 days in the presence of 10 ng ml$^{-1}$ M-CSF (R&D Systems)[36].

**Patients included in this study.** All patients were positive for SARS-CoV2 by PCR from throat swab and admitted to the ICU and receiving ventilatory support due to severe pneumonia with a component of acute respiratory distress syndrome. Patients were two male and two female with age range 27–43 years admitted to ICU between 3 and 14 days.

**Air-liquid interface epithelium model.** Primary nasal cells were isolated using a nasal brush (Dent-O-Care, #620B) inserted into the nasal turbinations and twisted. Cells were isolated from the brush by gently expelling monolayer medium (Airway Epithelial Cells Basal Medium, PromoCell, #C-21260 + 1 pack of Airway Epithelial Cell Growth Medium Supplement, PromoCell, #39160 + 100 U ml$^{-1}$ Penicillin/ Streptomycin, Gibco #10378) and PBS to wash of cells. Cells were cultured in monolayer culture in tissue culture flask (Sarstedt, TC: Standart #83.3911) coated with 0.1 mg ml$^{-1}$ Bovine type I collagen solution (Sigma-Aldrich, #804592, diluted in sterile ddH$_2$O). Monolayer cultures were split using 1x Trypsin mixed with 0.3 mM EDTA (10x Trypsin (2.5%), Gibco, #15090, diluted to working concentration in PBS + UltraPure 0.5 M EDTA, Invitrogen, #15575) at ~80% confluency. At passage two, cells were seeded at 2–3 × 10^4 cells on 6,5 mm Transwell membranes (Corning, #3470) coated with 30 ug/ml Bovine type I collagen solution (Sigma-Aldrich, #804592, diluted in sterile ddH$_2$O). Cells were seeded and submerged in 2x P/S (200 U ml$^{-1}$ Pen/Strep) DMEM-low glucose (Sigma-Aldrich, D5921) mixed one to one with 2x Monolayer medium (Airway Epithelium Cell Basal Medium, (PromoCell, #C-21260) supplemented with 2 packs of Airway Epithelial Cell Growth Medium Supplement (PromoCell, #C-39160) without triiodothyronine + 1 ml of 1.5 mg ml$^{-1}$ BSA). When cultures reach full confluency ALI ( = Air-liquid interface) is introduced and medium is changed to ALI medium (Pneumacult ALI medium kit (StemCell, #5001) with ALI medium supplement (StemCell, #5001) and 100 U ml$^{-1}$ Pen/strep) supplemented with 24 ug of hydrocortisone (StemCell, #07925) and 0.2 mg heparin (StemCell, #07980). Membranes was allowed at least 21 days of differentiation verified by extensive cilia beating and mucus covering.

Upon initiation of treatment ALI cultures was washed for 5 min using DMEM (low glycose, no additives) and baso-lateral medium changed for ALI medium containing either 150 uM 4-OI or DMSO. Baso-lateral medium containing treatment was left overnight. A 100 ul of DMEM (low glycose) with 150 uM or DMSO was additionally added to the apical compartment overnight. At time of infection, apical medium was removed and 100 ul DMEM (low glycose) containing SARS-CoV-2 at MOI 0.1 was added to all membranes for 1 h and placed in 37 °C incubator. After 1-h apical infection medium was removed and membranes placed in 37 °C incubator for 24 h before harvest.

At time of harvest, baso-lateral medium was removed and 500 ul Trypsin/ EDTA was added bao-laterally and 200 ul was added apically. After ~5 min cells were harvested using 5% FPB/DMEM (low glycose). Cells were lysed for RNA isolation using lysisbuffer from High Pure RNA Isolation Kit (Roche Diagnostics, #11828665001). For Western blot cells were lysed in RIPA buffer containing 1/10 Protease inhibitor (Roche), 1/1000 Benzonase (Sigma, #E1014) and 1/50 0.5 M Sodium Flouride.

**Short-interfering RNA (siRNA)-mediated knock down.** For short interfering RNA experiments, HaCat cells were transfected in 6-well plates with 80 pmol of human Nrf2(1) (sc-37030) or control si RNA (sc-37007) diluted in serum and antibiotic free DMEM and using Lipofectamine RNAi Max as per manufacturer's instructions. HaCat cells were incubated for 72 h in the presence of the siRNA before being processed. For experiments in Calu-3 cells, the same protocol conditions were used with control si RNA (sc-37007), Keap1 siRNA (sc-43878) and HO-1 (sc-35554) that were lipofected for 48 h.

**dsDNA, cGAMP and optimized RIG-I agonist stimulation of cells.** HSV-60 naked, a viral dsDNA motif, 2'3'-cGAMP, a STING ligand, and M8, a sequence optimized RIG-I agonist[37] were obtained from Invivogen and John Hiscott (Pasteur Institute, Rome), respectively. Intracellular delivery of dsDNA and cGAMP was achieved using Lipofectamine 2000 (Invitrogen) diluted in serum-free medium with a ratio of Lipo.dsDNA/cGAMP of 1:1. Final concentration for both dsDNA and cGAMP was 4µg.mL$^{-1}$. Intracellular delivery of M8 was achieved using Lipofectamine RNAiMax (Invitrogen) diluted in serum-free medium with a ratio of Lipo.RNA of 1:1. Final concentration of M8 was 10 ng.mL$^{-1}$.

**VACV infection assays.** BMDCs, HaCaT and HaCaT Stat1 KO were incubated or not with 150 µM 4-OI for 48 h before infection with vtag2GFP using 0.1 or 0.01 pfu per cell at 37 °C for 60 min. Then, infected cells were washed to remove potential unbound viruses and infection proceeded at 37 °C.

To determine the proportion of vtag2GFP infected cells GFP expression was detected at 16 hpi by flow cytometry using triplicates. Briefly, cells were harvested, washed with FACS buffer (PBS, 0.01% sodium azide, and 0.1% BSA) and fixed with paraformaldehyde 4% in PBS for 10 min. After extensive washing with FACS buffer, $2 \times 10^4$ cells were scored and analyzed in a FACSCalibur flow cytometer (BD Sciences) per experimental condition in triplicates. To determine VACV virus

titres, HaCaT and HaCaT STAT1 KO cells were previously stimulated for 48 h or not with 150 μM 4-OI and then infected with VACV-WR using 0.1 or 0.01 pfu per cell at 37 °C. At 24 hpi, cells were harvested in their own media, centrifuged at 1800 × g for 5 min, and resuspended in 0.5 ml of fresh medium. In all cases, samples were frozen, thawed three times and titrated using duplicates in BSC-1 cells. Briefly, preconfluent monolayers of BSC-1 cells were infected with tenfold serial dilutions of viral inoculums for 1 h at 37 °C. Then, inoculum was replaced with semi-solid carboxy-methyl cellulose (Sigma) media with 2% FBS and cells fixed in 10% formaldehyde at 3 dpi. Plaques were stained with 0.1% (w/v) crystal-violet. Two independent experiments were performed.

**Zika virus infections**. A549 cells (kind gift from G. Kochs, Freiburg) and Huh-7 were cultured at 37 °C in DMEM, supplemented with 10% FCS and 2mM L-Glutamine. Cells were seeded in 24 well plates and pre-treated with 4-octyl-itaconate (4-OI) (150 uM) for 48 h. Cells were infected with ZIKV (moi 0.1) for 1 h. 4-OI was freshly added when the medium was changed.Cells were lysed and total RNA was extracted at 96hpi using the QIAshredder (Qiagen) and RNeasy Mini Kit (Qiagen) according to the manufacturer's instructions. RNA was reverse transcribed using SuperScript II Reverse Transcriptase (Invitrogen) into cDNA that was then used for qPCR with SYBR green PCR kit (Life Technologies). $C_T$ values were normalized to GAPDH ($\Delta C_T$). SYBR green primer probes used include GAPDH (for: CATGGCCTTCCGTGTTCCTA, rev: CCTGCTTCACCACCTTCTTGA) and ZIKV (for: CGAGGAACATCCAGACTC, rev: ATTGGAGATCCTGAAGTTCC).

**SARS-CoV2 TCDI50% assay**. The assay was performed as follows. 2 × 104 Vero E6 TMPRSS2 cells were seeded in 90 ul DMEM (Gibco, + 2% FCS (Sigma-aldrich) + 1% Pen/Strep (Gibco) + L-Glutamine (Sigma-Aldrich) per well in flat-bottom 96-well plates. 24 h after, samples were titrated onto the cells by addition of 10ul of a 10-fold serial dilution. One full plate was used per sample analyzed. Each dilution of supernatant were represented 8 times on a plate. The cells were incubated for 72 h in a humidified $CO_2$ incubator at 37 ˚C, 5% $CO_2$, before fixing with 5% Formalin (Sigma-Aldrich) and staining with crystal violet solution (Sigma-Aldrich). Images were taken using a Leica DMi1, microscope with a Leica MC170 HD camera. TCDI50 % virus titer calculated by Reed-Muench method. SARS-CoV2 primers and probes used were: Forward primer: AAATTTTGGGGACC AGGAAC, Reverse primer TGGCACCTGTGTAGGTCAAC, and probe FAM-AT GTCGCGCATTGGCATGGA-BHQ.

**Western blot**. Cells were lysed in 100 μL of ice-cold Pierce RIPA lysis buffer (Thermo Scientific) supplemented with 10 mM NaF, 1x complete protease cocktail inhibitor (Roche) and 5 IU.mL$^{-1}$ benzonase (Sigma), respectively. Protein concentration was determined using a BCA protein assay kit (Thermo Scientific). Whole-cell lysates were denatured for 3 min at 95°C in presence of 1x XT Sample Buffer (BioRad) and 1x XT reducing agent (BioRad). In total, 10–40 μg of reduced samples were separated by SDS-PAGE on 4–20% Criterion TGX precast gradient gels (BioRad). Each gel was run initially for 15 min at 70 V and 45 min at 120 V. Transfer onto PVDF membranes (BioRad) was done using a Trans-Blot Turbo Transfer system for 7 min. Membranes were blocked for 1 h with 5% skim-milk (Sigma Aldrich) at room temperature in PBS supplemented with 0.05% Tween-20 (PBST). Membranes were fractionated in smaller pieces and probed overnight at 4 °C with any of the following specific primary antibodies in PBST: anti-Nrf2 (12721, Cell Signaling 1:1000), anti-TBK1/NAK (3013, Cell Signaling 1:1000), anti-phospho-TBK1/NAK (5483, Cell Signaling 1:1000), anti-SQSTM1/p62 (8025, Cell Signaling 1:1000), anti-IRF3 (11904, Cell Signaling 1:1000), anti-phospho-IRF3 (4947, Cell Signaling 1:500), anti-HO-1 (5853, Cell Signaling 1:1000), anti-IFIT1 (14769, Cell Signaling 1:1000), anti-NRF2 (12721, Cell Signaling 1:1000), anti-STING (13647, Cell Signaling 1:1000), anti-NqO1 (3187, Cell Signaling 1:1000), anti-STAT1 (9172, Cell signaling, 1:1000), SARS-CoV2 spike antibody (GeneTex, cat# GTX632604, 1:1000), and anti-Vinculin (18799, Cell Signaling 1:1000) used as loading control. After three washes in PBST, secondary antibodies, peroxidase-conjugated F(ab)2 donkey anti-mouse IgG (H + L) (1:10000) or peroxidase-conjugated F(ab)2 donkey anti-rabbit IgG (H + L) (1:10,000) (Jackson ImmunoResearch) were added to the membrane in PBST 1% milk for 1 h at room temperature. All membranes were washed three times and exposed using either the SuperSignal West Pico PLUS chemiluminescent substrate or the SuperSignal West Femto maximum sensitivity substrate (ThermoScientific) and an Image Quant LAS4000 mini imager (GE Healthcare).

**Semi-native WB dimerization assay**. IRF3 dimerization was assayed under semi-native conditions. Cells were lysed in ice-cold Pierce RIPA lysis buffer (Thermo Scientific) supplemented with 10 mM NaF, 1x protease cocktail inhibitor (Roche) and 5 IU mL$^{-1}$ benzonase (Sigma). Protein concentration was determined using a BCA protein assay kit (Thermo Scientific). Whole-cell lysates were mixed with 1x XT Sample Buffer (BioRad); samples were neither reduced nor heated before separation was done on 4–20% Criterion TGX precast gradient gels (BioRad) by SDS-PAGE electrophoresis. Each gel was run initially for 15 min at 70 V and 15 min at 120 V. Transfer onto PVDF membranes (BioRad) was done using a Trans-Blot Turbo Transfer system for 7 min. Membranes were blocked for 1 h with 5% skim-milk (Sigma Aldrich) at room temperature in PBS supplemented

with 0.05% Tween-20 (PBST). Membranes were probed overnight at 4 °C with the following specific primary antibody in PBST: anti-IRF3. After three washes in PBST, secondary antibodies, peroxidase-conjugated F(ab)2 donkey anti-rabbit IgG (H + L) (1:10,000) (Jackson Immuno Research) were added to the membrane in PBST 1% milk for 1 h at room temperature. All membranes were washed three times and exposed using either the SuperSignal West Pico PLUS chemiluminescent substrate or the SuperSignal West Femto maximum sensitivity substrate (Thermo Scientific).

**Cytokine qPCR analysis**. Gene expression was determined by real-time quantitative PCR, using TaqMan detection systems (Applied Biosciences). RNA was extracted using the High Pure RNA Isolation kit (Roche) and RNA quality was assessed by Nanodrop spectrometry (Thermo Fisher). RNA levels were analyzed using premade TaqMan assays and the RNA-to-Ct-1-Step kit according to the manufacturer's recommendations (Applied Biosciences). Taqman assays for qPCR were purchased from Applied Bioscience; IFNB1 (Hs01077958), CXCL10 (Hs00171042), CCL5(Hs00982282), IL1B (Hs01555410), HMOX1 (Hs01110250), ACTB (Hs01060665), and TNFA (Hs00174128).

**Transcriptome analysis COVID19**. COVID19 data set analysis (Fig. 1): RNA-seq data were obtained from an already available dataset from Blanco-Melo (doi: [https://doi.org/10.1101/2020.03.24.004655]). From the raw read-counts differential expression values were calculated using DESeq2[38]. Significantly differentially expressed genes (SDEGs) were selected based on the thresholds of adjusted p value < 0.05 and absolute fold change of 2 (Fig. 1a). In order to focus on commonality across the different conditions with respect to generating clinically relevant hypotheses, the 815 SDEGs obtained from the biopsy sample were checked if these were also present in the list of SDEGs in the other conditions. All genes that occur in at least 3 or more other conditions were included in the final list, resulting in 113 genes. Finally, the expression values for all genes in the final list across all conditions were assembled, clustered using Euclidean distance metric and Ward's variance minimization algorithm, and visualized as a heatmap using Python3.7 and seaborn cluster-map tools. Then the gene-sets from each of the outlined clusters were used for pathway enrichment analysis using Enrichr (Fig. 1b)[39]. Finally, the STRING database[40] was used to construct the cloud network starting from lists of genes manually annotated for NRF2, inflammation and IFN signaling. Edges and nodes were extracted from STRING and imported to Cytoscape[41] version 3.7 for further visualization (Fig. 1c).

**Transcriptome analysis of HSV1-infected HaCaT cells**. RNA sequencing was performed in collaboration with BGI Europe Genome Center (Copenhagen, Denmark) following the standard operational procedure as described before[42]. Briefly, the quality of total RNA was checked using the Agilent 2100 bioanalyzer. To construct the sequencing library for MGIseq-2000, ~1 μg of polyA enriched RNA was used for library construction using the MGIEasy RNA Directional Library Prep Kit (MGI Tech). Next, paired-end sequencing with 100 cycles was performed using the MGISEQ-2000 sequencing instrument, according to the manufacturer's instructions. We generated an average of 63 million raw reads for each sample. The clean RNA reads were first aligned to the hg19 UCSC RefSeq (RNA sequences, GRCh37) using bowtie2 at first. To map the transcripts from the viruses, the unmapped reads were then aligned to the coding sequence of the human herpesvirus 1 (KOS strain). The expression of human genes and virus genes were performed by transforming mapped transcript reads to TPM using RSEM[43]. The normalized expression were estimated and normalized by DEseq2. Differentially expressed genes were defined as genes with fold change over twofolds and adjusted p value < 0.001 using DESeq2.

**Cell painting assay**. The procedure from Bray et al.[44] was followed with adaption to 96-well plates and optimized fluorophore concentrations for Vero E6 cells and the Celldiscoverer 7 imaging system:

5000 wt Vero E6 cells were seeded into the inner 60 wells of a 96-well plate with optical bottom (Corning #3603) in 75 μL full growth medium. After 24 h, compounds were dosed as a 25 μL 4X solution (final DMSO = 0.5%). After 24 h, 75 μL medium was removed and replaced with 75 μL medium containing 500 nM MitoTracker Deep Red (final C = 325 nM) and plates were incubated (37 °C, 5% CO2, humid) in the dark for 30 min. Wells were then aspirated and 75 μL medium were added, before adding 25 μL 16% paraformaldehyde (Electron Microscopy Sciences, cat. no.15710- S) (final PFA = 4%) and incubating in the dark for 20 min. Plates were washed once with 1X HBSS (Invitrogen, #14065-056) and 75 μL 0.1% (vol/vol) Triton X-100 (BDH, #306324 N) in 1X HBSS was added and incubated for 15 min in the dark. Plates were washed twice with 1X HBSS before addition of 75 μL multiplex staining solution (final concentrations: 0.75 μg mL$^{-1}$ WGA-AF555; 8.75 μg mL$^{-1}$ Concanavalin-AF488; 2.5 μL mL$^{-1}$ Phalloidin-AF568; 0.75 μM SYTO 14; 5 μg mL$^{-1}$ Hoechst 33342 in 1% (wt/vol) bovine serum albumin (Sigma-Aldrich, #A9647)) and incubation for 30 min in the dark. Plates were then washed three times with 1X HBSS, with no final aspiration and imaged immediately in a Zeiss Celldiscoverer 7 automated microscope. 9 images are acquired in each well with 2×2 binning using the AxioCam 702 CMOS 12-bit camera with 4x analog gain. To generate the bioactivity profiles the workflow

outlined in Svenningsen & Poulsen[45] was followed. In short, CellProfiler 2.1.1[46] was used to correct images for uneven illumination followed by image segmentation and extraction of 1476 features across nuclei, cytoplasm, and the whole cell on a per-cell basis. Features were then averaged to per-well profiles after which the data was normalized on a per-plate basis followed by per-treatment aggregation which affords the final profiles. The heatmap of morphological profiles is visualized with heatmap.2 in the gplots package. The Pearson correlation matrix is calculated using the stats package in R 3.6.0 (R core Team) and visualized using the corrplot 0.84 package (R package corrplot).

Hierarchical clustering of the correlation matrix is performed using the stats package with Pearson correlation coefficients as distance metric and average linkage method. To determine activity scores and thresholds the workflow described by Hutz et al.[47] was followed.

**Ethics**. The project was approved by Institutional review boards at Aarhus University Hospital, by the Danish National Committee in Health Research Ethics (1-10-72-80-20) and the Danish Data protection Agency in accordance with the ethical standards of the Helsinki Declaration. Written informed consent was obtained from all study participants.

**Reporting summary**. Further information on research design is available in the Nature Research Reporting Summary linked to this article.

## Data availability

For COVID19 data set analysis (Fig. 1) the RNA-seq data were obtained from an already available dataset from Blanco-Melo et al. ([https://doi.org/10.1101/2020.03.24.004655])[14]. Transcriptome analysis of lung autopsies obtained from five individual COVID-19 patients (Desai et al.[15]) [https://doi.org/10.1101/2020.07.30.20165241] GEO accession code GSE150316. For RNA sequencing, files generated for analysis of HaCaT cells infected with HSV in the presence or absence of 4-OI have been deposited to the data depository database (CNGBdb, [https://db.cngb.org]) with the following accession number: CNP0001039. Source data are provided with this paper.

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

## Acknowledgements

This research work was supported by Ester M og Konrad Kristian Sigurdssons Dyreværnsfond, Beckett-Fonden, Kong Christian IX og Dronning Louises Jubilæumslegat, Læge Sofus Carl Emil Friis og Hustru Olga Doris Friis´ legat, Købmand I Odense Johan og Hanne Weimann Født Seedorffs Legat, Hørslev Fonden, UK Medical Research Council (MRC core funding of the MRC Human Immunology Unit; JR), Lundbeck foundation (R303-2018-3379 and R219-2016-878, and R268-2016-3927), and Independent Research Fund Denmark – Medical Sciences (9039-00078B, 4004-00047B, and 0214-00001B). CarlsbergFoundation (Semper Ardens) and European Research Council (ERC-AdG ENVISION; 786602). Marie Skłodowska-Curie Action of the European Commission # 813343 and Italian Cancer Research Society #22891 to JH.

## Author contributions

D.O. and C.H. conceived the project and prepared figures. D.O., A.L.H., J.T., J.C., B.H., M.I., E.B.S., A.K., L.R., M.I., M.S., S.F., M.T., M.L., H.H., V.G., A.H., A.H., A.K., C.G., D.V.D.H., C.M., L.B., A.T., T.A., J.H., R.O., S.E.J., C.G.N., L.B., Y.L., and A.A. performed experiments, analyzed data, and prepared figures. E.F. performed bioinformatics and prepared figures. T.M. was responsible for including COVID-19 patients and achieving patient material. T.A., J.R., A.A., T.M., S.R.P., and C.H. planned experiments and analyzed data. T.P. was responsible for 4-OI synthesis. C.H. and D.O. drafted and finalized the manuscript. J.H. planned experiments and edited the manuscript. E.F., S.P., S.B., and M.L. performed bioinformatics and prepared figures. M.J. facilitated SARS-CoV2 laboratories.

## Competing interests

The authors declare no competing interests.

## Additional information

David Olagnier [1,15✉], Ensieh Farahani[1,14], Jacob Thyrsted[1,14], Julia Blay-Cadanet[1,14], Angela Herengt[1,14], Manja Idorn [1], Alon Hait[1,2], Bruno Hernaez[3], Alice Knudsen[1], Marie Beck Iversen[1], Mirjam Schilling[4], Sofie E. Jørgensen[1,2], Michelle Thomsen[1,2], Line S. Reinert [1], Michael Lappe[5], Huy-Dung Hoang[6], Victoria H. Gilchrist[6], Anne Louise Hansen[1], Rasmus Ottosen[7], Camilla G. Nielsen[1], Charlotte Møller[1], Demi van der Horst[1], Suraj Peri[8], Siddharth Balachandran[8], Jinrong Huang[9,10], Martin Jakobsen [1], Esben B. Svenningsen [7], Thomas B. Poulsen[7], Lydia Bartsch[11], Anne L. Thielke[1], Yonglun Luo [1,9], Tommy Alain[6], Jan Rehwinkel [4], Antonio Alcamí [3], John Hiscott[12], Trine Mogensen[1,2,13], Søren R. Paludan[1] & Christian K. Holm [1,15✉]

[1]Department of Biomedicine, Aarhus Research Center for Innate Immunology, Aarhus University, Aarhus, Denmark. [2]Department of Infectious Diseases, Aarhus University Hospital, Aarhus, Denmark. [3]Centro de Biología Molecular Severo Ochoa (Consejo Superior de Investigaciones Científicas - Universidad Autónoma de Madrid), Nicolás Cabrera 1, 28049 Madrid, Spain. [4]Medical Research Council Human Immunology Unit, Medical Research Council Weatherall Institute of Molecular Medicine, Radcliffe Department of Medicine, University of Oxford, Oxford OX3 9DS, UK. [5]Omiics ApS, Åbogade 15, 8200 Aarhus N, Denmark. [6]Children's Hospital of Eastern Ontario Research Institute, Department of Biochemistry Microbiology and Immunology, University of Ottawa, Ottawa, ON K1H 8L1, Canada. [7]Department of Chemistry, Aarhus University, Aarhus, Denmark. [8]Fox Chase Cancer Center, 333 Cottman Avenue, Philidelphia, PA 19111-2497, USA. [9]Lars Bolund Institute of Regenerative Medicine, BGI-Shenzhen, Shenzhen 518083, China. [10]Department of Biology, University of Copenhagen, 2100 Copenhagen, Denmark. [11]Department of Pediatrics and Adolescent Medicine, Division of Pediatric Neurology, University Medical Center Göttingen, 37075 Göttingen, Germany. [12]Istituto Pasteur Italia-Cenci Bolognetti Foundation, Viale Regina Elena 291, 00161 Rome, Italy. [13]Department of Clinical Medicine, Aarhus University, Aarhus, Denmark. [14]These authors contributed equally: Ensieh Farahani, Jacob Thyrsted, Julia B. Cadanet, Angela Herengt. [15]These authors jointly supervised this work: David Olagnier, Christian K. Holm. ✉email: olagnier@biomed.au.dk; holm@biomed.au.dk

