## [Peer Review File · Nature Communications]

Reviewers' Comments:

Reviewer #1:

Remarks to the Author:

Review of Olnagier et al

In this study, the authors describe 4-octyl-itaconate (4-OI) and dimethyl fumarate (DMF) as potential antiviral and anti-inflammatory agents against RNA and DNA viruses including SARS-CoV-2, VACV, HSV-1 and ZIKV. The identification of novel classes of antiviral agents (with additional protective activity) is certainly an important goal, especially against SARS-CoV-2 given the ongoing pandemic. Initially, the authors performed transcriptional analysis on SARS-CoV-2 biopsies and identified genes that were downregulated in the Nrf2 signaling pathway. They subsequently perform a series of in vitro assays to show that the antiviral activity of 4-OI against SARS-CoV-2 correlates with NRF2 expression. The authors also demonstrate that the antiviral and anti-inflammatory activity of 4-OI is not limited to SARS-CoV-2 but extends to other viruses. Mechanistically, the authors suggest that the antiviral activity of 4-OI against SARS-CoV-2 is NRF2-dependent and type I IFN-independent, although how this occurs is not entirely clear. In addition, the authors suggest that the metabolite DMF also promotes a pan antiviral activity against different human viruses.

The strengths of the paper include the novelty, the identification of possible broad-spectrum antiviral and anti-inflammatory compounds with in vitro activity against SARS-CoV-2, the use of multiple cells (including primary PBMCs and human airway epithelium cultures), and the impressive magnitude of antiviral effects in some of the viral yield assays. A possible combination antiviral/anti-inflammatory is attractive for many types of viral infections including SARS-CoV-2. The weaknesses of the paper include inadequate cytotoxicity analysis, a lack of statistical rigor throughout, weak in vivo data, and an absence of a clear mechanism for how 4-OI and DMF (and NRF2) confer antiviral effects. Thus, further experiments and analysis are required to buttress their findings.

Major Issues

1. Cytotoxicity issues. The authors need to perform more detailed cytotoxicity analyses with each of the cells that they perform 4-OI and DMF treatments and viral function. Ideally, a CC50 curve (MTT-based assay) will be established so the reader can assess how much of the antiviral effect is likely due to effects on cell metabolism. Currently, they performed a single LDH-based cytotoxicity assay in Vero cells at one concentration of 4-OI (Fig 2g).

2. Linkage of phenotype to NRF2. (a) The authors make a significant point that 4-OI and DMF function as antiviral agents by stabilizing and activating NRF2. However, the only data supporting this is a single siRNA experiment in Fig 3f-g, and here the phenotype is quite modest. To confirm this key aspect of the paper, the authors should edit NRF2 expression more completely by CRISPR (as was published in PMID:28289076) and repeat the 4-OI and DMF treatment and viral infection experiments. (b) Beyond this, the authors never confirm the NRF2-dependence of the antiviral effects against other viruses (beyond HSV) in Figure 3: Finally, 4-OI treatment seems to have a proviral effect on VSV and the authors fail to explain why a RNA virus from a different family evokes an opposite response.

3. Mechanism of broad spectrum antiviral activity. The authors suggest that the broad-spectrum antiviral effects of 4-OI and DMF are independent of type I IFN based on experiments in Vero cells (which lack IFN genes) as well as STAT1 KO cells. However, no mechanism is actually provided for this antiviral activity either at the host or viral level. The paper would be strengthened by defining one of these aspects. Absent this, the authors must provide in the Discussion a more scholarly assessment of the possible nature and mechanism of such a broad spectrum antiviral agent that affects RNA and

DNA viruses. Have they performed RNAseq in cells treated with 4-OI or DMF (ideally in WT and NRF2 KO cells)?

4. Rigor. In many of the Figure legends, the authors indicate only one single experiment was performed. This is not adequate, as all data must be reproduced at least once in a separate independent biological experiment. Moreover, in many of these experiments with only one biological replicate, the authors perform statistical analysis, which is not appropriate. This needs to be addressed in multiple Figures. Indeed, no statistical analysis was performed in Figures S1, S2, S3, S4, S5, 2b, 2d, 2j, 2l, 2p, 2q, 3h, 3i, 3j, 3l and all panels of Figure 4.

5. In vivo data. The in vivo data with HSV in Fig S4 has several issues that must be addressed: (a) The Figure is lacking statistical analysis; (b) In the middle panel, the survival data differences are not particularly compelling. At best, there is a one to two-day difference in survival; (c) The legend is unclear. Is the data pooled from two experiments or representative of a single experiment? The data should be pooled together; (d) There is no WT control for survival after treatment with 4-OI. This is needed; (e) The authors make a statement on line 220 that "pre-treatment with 4-OI decreased disease progression, an effect that was enhanced in mice deficient in STING (TMEM173-/-)". How do they know the effect was enhanced in STING-deficient mice compared to WT mice without any direct statistical comparison? and (f) no viral burden data in vivo is provided to support their proposed mechanism of action.

Minor Issues.

1. Line 97. This statement is written awkwardly, since NRF2 inhibits IFN and STING signaling. Thus, it would not be expected to have an antiviral effect. Indeed, that is one of the unexpected findings of the paper.

2. Line 121-123. There is no direct data that SARS-CoV-2 targets the NRF2 pathway. Rather, the data are correlative. This statement should be edited.

3. Line 133. "Deferentially" should be "differentially."

4. Line 157. Fig 2l and m do not show viral RNA levels as suggested by the text.

5. Figure 2d, j, and l. What do these panels actually measure? The y-axis indicates "cytopathogenic effect." Where do these numbers come from? TCID50 assays? More detail is needed in the legend to explain what assay this is.

6. Lines 228-231. Why not just CRISPR IFNAR1 or treat with a commercially available blocking anti-IFNAR1 mAb?

7. Line 262. Abolished is not correct. There still is expression of some of the cytokines and chemokines.

8. Mechanism of IRF3 inhibition by 4-OI. The authors show data on the loss of IRF3 dimerization but never speculate as to how 4-OI and NRF2 signaling could cause this to happen?

9. Lines 342-343. The statement on NRF2 deficiency in humans seems premature and speculative and likely should be deleted.

10. Fig 2a has a reduction of viral RNA to 6 log₁₀ with 125 uM of 4-OI but the same dose in Fig 2b has

a reduction to $3\log_{10}$. What is the basis for the 1,000-fold discrepancy?

Reviewer #2:

Remarks to the Author:

In this paper, Olagnier et. al. show that activation of NRF2, an oxidative stress responder, can inhibit replication of both RNA and DNA viruses including SARS-CoV2. The authors test multiple viruses in human and non-human cell lines. They also show that siRNA-mediated knockdown of NRF2 results in increased viral replication while treatment of cells with NRF2 agonists 4-OI and DMF result in reduced viral replication. This antiviral activity is seen in the absence of STAT1 signaling suggesting that it is IFN-independent. However, the authors also present convincing data that NRF2 activation can suppress production of inflammatory cytokines including CXCL10. Given the strong correlation between CXCL10 and viral load in rhinovirus and SARS-CoV2 infections (Cheermala, 2020 MedRxiv, Landry, 2018, JID), the data showing 4-OI mediated reduction of CXCL10 in patient blood samples is exciting

This is an important contribution to the field of NRF2 antiviral activity and potential SARS CoV2 antivirals. I would recommend acceptance with minor revisions. I think the authors need to address some previous findings on NRF2 antiviral activity and restate some findings in the text. Specifically, the authors make a connection between NRF2-mediated suppression of IRF3 dimerization and reduction in inflammatory cytokine production and suppression of RIG-I signaling. However, this could also occur via other pathways including NFkB signaling which is not tested. MAVS is also shown to be suppressed by NRF2 in the absence of RIG-I activation which could also explain the suppression of RIG-I and STING signaling. In the absence of experiments targeting IRF3 activation (dimerization or phosphorylation) and showing subsequent reduction in inflammatory cytokine signaling, these overstatements should be modified (examples include lines 107-108, 294-295 etc.).

Minor points:

- 1) Mihaylova et. al. 2018 showed that NRF2 antagonizes RIG-I activation in rhinovirus infection and that basal NRF2 activation varies in nasal epithelia vs. basal bronchial epithelial cells. Since these cells are also infected by SARS CoV2 discussion of these findings is highly relevant in this paper.
- 2) In Figure 1b, there is a set of genes activated in SARS-CoV2 biopsy and suppressed in all the cell lines – are RIG-I/ TLR signaling in this cluster? Also, the three dotted lines suggest three distinct clusters which are not highlighted in the text list of the genes.
- 3) Diverging colour palette should have its center set at zero, please recolour figures 1b-c.
- 4) Please redraw Figure 1d so the gene names are readable. If the intent is to not have them be readable please remove the text.
- 5) Please show survival data for WT mice treated with 4-OI prior to HSV-2 infection (Figure S4).
- 6) Please show individual data points in all the bar plots (example Figures 3h-j, 3l), for reference see Weissberger et. al., 2015, PloS Biology.
- 7) In Figure 3g, comparing NRF2 silenced+4-OI treated vs. 4-OI treated alone suggests that NRF2-independent pathways may also be responsible for 4-OI mediated antiviral activity. Is this difference significant?
- 8) In general, statistics are missing from multiple figures – for experiments that contain at least 3 replicates, statistics should be shown (see Figures 4a-d and others).
- 9) The legend for Figure 4d-f says that 'data are representative of one experiment performed in duplicates where means and s.e.m. are displayed.'. Does this mean that this experiment was only done once? If so, it needs to be repeated. If this is a representative experiment of multiple independent experiments then please restate the text.
- 10) Why are Figure 4g,h separate?

- 11) Please label cell type in western blot experiments (e.g. Figure 4n,o,q,t)
- 12) The figure axes and microscopy labels in Figure S2 are very hard to read, please resize.

Reviewer #1 (Remarks to the Author)

Review of Olganier *et al.*

In this study, the authors describe 4-octyl-itaconate (4-OI) and dimethyl fumarate (DMF) as potential antiviral and anti-inflammatory agents against RNA and DNA viruses including SARS-CoV-2, VACV, HSV-1 and ZIKV. The identification of novel classes of antiviral agents (with additional protective activity) is certainly an important goal, especially against SARS-CoV-2 given the ongoing pandemic. Initially, the authors performed transcriptional analysis on SARS-CoV-2 biopsies and identified genes that were downregulated in the Nrf2 signaling pathway. They subsequently perform a series of *in vitro* assays to show that the antiviral activity of 4-OI against SARS-CoV-2 correlates with NRF2 expression. The authors also demonstrate that the antiviral and anti-inflammatory activity of 4-OI is not limited to SARS-CoV-2 but extends to other viruses. Mechanistically, the authors suggest that the antiviral activity of 4-OI against SARS-CoV-2 is NRF2-dependent and type I IFN-independent, although how this occurs is not entirely clear. In addition, the authors suggest that the metabolite DMF also promotes a pan antiviral activity against different human viruses.

The strengths of the paper include the novelty, the identification of possible broad-spectrum antiviral and anti-inflammatory compounds with *in vitro* activity against SARS-CoV-2, the use of multiple cells (including primary PBMCs and human airway epithelium cultures), and the impressive magnitude of antiviral effects in some of the viral yield assays. A possible combination antiviral/anti-inflammatory is attractive for many types of viral infections including SARS-CoV-2. The weaknesses of the paper include inadequate cytotoxicity analysis, a lack of statistical rigor throughout, weak *in vivo* data, and an absence of a clear mechanism for how 4-OI and DMF (and NRF2) confer antiviral effects. Thus, further experiments and analysis are required to buttress their findings.

We would like to thank Reviewer 1 from his/her critical assessment of our manuscript. We were pleased with the overall positive nature of his/her comments especially emphasizing on the novelty of the broad-spectrum antiviral activity of the molecules discovered and on the impressive magnitude of the antiviral effect observed against different viruses. However, this reviewer also raised some major and minor concerns that we are addressing in the point-by-point response below.

Major Issues

1. Cytotoxicity issues. The authors need to perform more detailed cytotoxicity analyses with each of the cells that they perform 4-OI and DMF treatments and viral function. Ideally, a CC50 curve (MTT-based assay) will be established so the reader can assess how much of the antiviral effect is likely due to effects on cell metabolism. Currently, they performed a single LDH-based cytotoxicity assay in Vero cells at one concentration of 4-OI (Fig 2g).

This is a highly valid point. We have addressed the comment above by performing detailed cytotoxicity analyses using LDH release assay with each of the cell lines used (HaCat, Calu-3 and Vero cells) and for each of the molecule tested (4-octyl-itaconate and dimethyl fumarate) and that for several doses. We also performed in parallel qPCR experiments of the *HMOX-1* gene. These analyses are placed in a new supplementary figure 2.

2. Linkage of phenotype to NRF2. (a) The authors make a significant point that 4-OI and DMF function as antiviral agents by stabilizing and activating NRF2. However, the only data supporting this a single siRNA experiment in Fig 3f-g, and here the phenotype is quite modest. To confirm this key aspect of the paper, the authors should edit NRF2 expression more completely by CRISPR (as was published in PMID:28289076) and repeat the 4-OI and DMF treatment and viral infection experiments. (b) Beyond this, the authors never confirm the NRF2-dependence of the antiviral effects against other viruses (beyond HSV)

We agree with this point from reviewer 1 that the only piece of data supporting a Nrf2 dependency of 4-OI antiviral activity is the siNrf2 experiment in HaCat cells infected with HSV-1. As described by us and others, (1) 4-OI is by far the most potent Nrf2 inducer and (2) Nrf2 is a difficult gene/protein to target for transient silencing/gene editing. As seen in Fig. 3f, the silencing of Nrf2 is only partial and cells still respond to 4-OI, as seen by the increase in HO-1 or Sqstm1 expression. This could explain the modest phenotype seen in Fig. 3g.

To address the points (a) and (b) above, we have tried to transiently silence Nrf2 either using a siRNA approach or a CRISPR/Cas9 approach in Calu-3 cells that are supportive for SARS-CoV-2 infection/replication (see below). Unfortunately, transient silencing of Nrf2 in Calu-3 cells did not impair its function as cells remained responsive to 4-OI as seen by the increase in HO-1 and NqO1 gene/protein levels. None of the treatments prevented 4-OI-mediated antiviral effect against SARS-CoV-2. Of note, Nrf2 inhibition using gRNA/Cas9 lead to a slight but significant increase in SARS-CoV-2 viral RNA.

Considering that 4-OI is an extremely potent Nrf2 inducer and that Nrf2 is difficult to silence in Calu-3 cells, we opted for another approach to demonstrate that Nrf2 induction leads to an antiviral program. Here, Calu-3 cells were silenced for the Nrf2 repressor Keap1 using siRNA. Interestingly, silencing of Keap1 lead to an approximately 1-log reduction in SARS-CoV-2 RNA levels and SARS viral titers. These data precipitated the generation of three new panels (s-u) in the revised new Figure 2, including the immunoblotting Fig. 2u which demonstrates that silencing of Keap1 leads to genetic

activation of Nrf2 pathway and to a reduction in SARS-CoV-2 infectivity. Altogether, we can convincingly say that at least some of the antiviral effect mediated by Nrf2 inducers is Nrf2 dependent. We have adjusted our text accordingly.

In Figure 3: Finally, 4-OI treatment seems to have a proviral effect on VSV and the authors fail to explain why an RNA virus from a different family evokes an opposite response.

VSV and especially VSVd51M used for this study is extremely sensitive to type I IFNs. 4-OI, as we previously showed in Olgagnier *et al*, 2018, Nature Communications and in Figure 4 of the current manuscript, is a strong suppressor of type I IFN responses. As VSVd51M is extremely sensitive to type I IFN, and could explain why 4-OI increases VSVd51M infectivity, rather than being antiviral as observed for other viruses such as SARS-CoV-2 and HSVs which have evolved to hijack the type I IFN response.

3. Mechanism of broad spectrum antiviral activity. The authors suggest that the broad-spectrum antiviral effects 4-OI and DMF are independent of type I IFN based on experiments in Vero cells (which lack IFN genes) as well as STAT1 KO cells. However, no mechanism is actually provided for this antiviral activity either at the host or viral level. The paper would be strengthened by defining one of these aspects. Absent this, the authors must provide in the Discussion a more scholarly assessment of the possible nature and mechanism of such a broad spectrum antiviral agent that affects RNA and DNA viruses. Have they performed RNAseq in cells treated with 4-OI or DMF (ideally in WT and NRF2 KO cells)?

This is a valid point by the reviewer. The KEAP1/NRF2 pathway regulates several hundred of genes. The anti-viral effect of 4-OI could be induced by one of these genes, or – even more likely – by a combination of these genes. To address this point in depth, we performed RNAseq analysis on 4-OI treated cells/infected with HSV-1 and used these data for pathway and gene analysis. This analysis is now presented in Fig S8, and identified several pathways that could be of importance to mediate the 4-OI antiviral effect.

Amongst the top up-regulated genes in response to 4-OI was the canonical Nrf2 responsive gene *HMOX-1* encoding for the HO-1 protein. Importantly, HO-1 was previously reported to have antiviral activity against viruses such as dengue and zika viruses. We thus logically tested whether HO-1 could be the Nrf2-target gene driving this antiviral program. The data are presented in a new Fig S9. Unfortunately, neither HO-1 overexpression nor HO-1 silencing was capable of altering SARS-Cov-2 replication in various cellular models.

Further, in an attempt to pin-point the anti-viral mode of action of 4-OI, we also used microscope-based analysis of morphology by cell-paint technology (Fig. S10). With this analysis we were able to compare morphological changes in cells treated with 4-OI to cells treated with compounds that have known cellular targets, and with cells treated with other compounds with reported anti-viral activity towards SARS-CoV2 including Remdesivir and Hydroxychloroquine. In this analysis, 4-OI was shown to have a low but significant morphological activity, without inducing a loss of cell viability. Interestingly, the activity of 4-OI did not overlap with other compounds with known perturbation in cell morphology including Rapamycin, Bafilomycin, Tunicamycin, Cyclohexamide, Emetine, Mitomycin, or Doxorubicin. Interestingly, there was also no observable overlap with the activity profile of Remdesivir or hydroxychloroquine, indicating that the anti-viral mode of action of 4-OI is distinct from other known anti-viral mechanisms.

4. Rigor. In many of the Figure legends, the authors indicate only one single experiment was performed. This is not adequate, as all data must be reproduced at least once in a separate independent biological experiment. Moreover, in many of these experiments with only one biological replicate, the authors perform statistical analysis, which is not appropriate. This needs to be addressed in multiple Figures. Indeed, no statistical analysis was performed in Figures S1, S2, S3, S4, S5, 2b, 2d, 2j, 2l, 2p, 2q, 3h, 3i, 3j, 3l and all panels of Figure 4.

This is a very valid point. Accordingly, we repeated all missing experiments so that all have been reproduced at least once. In addition, and to be even more thorough in terms of our rigor, we tested the effect of 4-OI on an additional primary SARS-CoV2 isolate from a patient in Japan, with the same result as with the isolate from Germany. Data with this new isolate is now placed in Fig. 2. Additionally, we have also re-analyzed another RNAseq data set, now including a total of 5 Covid-19 lung autopsies to demonstrate that Nrf2 pathway is suppressed in lungs from infected patients. This Figure replaces the original Fig. 1d of our manuscript.

We agree that statistical analysis could be applied more consistently. Accordingly, we added statistical analysis to panels 2b, 2p, 2q, 3h, 3i, 3j, 3l identified by the reviewer. However, a few of the panels identified by the reviewer are not eligible for re-analysis.

These are:

2d: This data is summarized in 2e, where the statistical analysis is performed.

2j: This data is summarized in 2k, where the statistical analysis is performed.

2l: This data is summarized in 2m, where the statistical analysis is performed.

5. In vivo data. The in vivo data with HSV in Fig S4 has several issues that must be addressed: (a) The Figure is lacking statistical analysis; (b) In the middle panel, the survival data differences are not particularly compelling. At best, there is a one to two-day difference in survival; (c) The legend is unclear. Is the data pooled from two experiments or representative of a single experiment? The data should be pooled together; (d) There is no WT control for survival after treatment with 4-OI. This is needed; (e) The authors make a statement on line 220 that “pre-treatment with 4-OI decreased disease progression, an effect that was enhanced in mice deficient in STING (TMEM173^{-/-})”. How do they know the effect was enhanced in STING-deficient mice compared to WT mice without any direct statistical comparison? and (f) no viral burden data in vivo is provided to support their proposed mechanism of action.

We agree with this point from Reviewer 1. Unfortunately, and because of the overall Covid-19 crisis, we were not able to access our animal facility or get new STING KO animals in the time frame to perform these revisions experiments. Since we were not able to repeat this *in vivo* experiment and because Reviewer 1 criticized this part of the work, we decided to remove this supplementary figure from our revised manuscript.

Minor Issues.

1. Line 97. This statement is written awkwardly, since NRF2 inhibits IFN and STING signaling. Thus, it would not be expected to have an antiviral effect. Indeed, that is one of the unexpected findings of the paper.

This statement has been adjusted accordingly in the revised version of our manuscript.

2. Line 121-123. There is no direct data that SARS-CoV-2 targets the NRF2 pathway. Rather, the data are correlative. This statement should be edited.

Indeed, our original submitted data were rather correlative and based on an RNA seq available data set from 1 Covid-19 patient. To increase the accuracy of our findings, we have re-analyzed another RNAseq data set, now including in total 5 Covid-19 lung autopsies to demonstrate that Nrf2 pathway is being suppressed in lungs from infected patients.

Furthermore, we also show that infection of Vero cells expressing human TMPRSS2 with SARS-CoV-2 for 48h leads to a suppression of Nrf2 protein levels and directly impacts the protein levels of canonical Nrf2-regulated proteins including HO-1 or NqO1. These findings are included as a new Fig.S1 in the revised version of our work.

3. Line 133. “Deferentially” should be “differentially.”

This has been edited appropriately in the revised document

4. Line 157. Fig 2l and m do not show viral RNA levels as suggested by the text.

This has been edited appropriately in the revised document

5. Figure 2d, j, and l. What do these panels actually measure? The y-axis indicates “cytopathogenic effect.” Where do these numbers come from? TCID50 assays? More detail is needed in the legend to explain what assay this is.

We agree with the reviewer that this could be more clearly stated. We have altered the legend accordingly.

6. Lines 228-231. Why not just CRISPR IFNAR1 or treat with a commercially available blocking anti-IFNAR1 mAb?

This is a valid point from Rev.1 and two additional panels using IFNAR KO cells have been added to the revised supplementary Figure 7.

7. Line 262. Abolished is not correct. There still is expression of some of the cytokines and chemokines.

This is correctly stated by the reviewer. We have changed the wording accordingly.

8. Mechanism of IRF3 inhibition by 4-OI. The authors show data on the loss of IRF3 dimerization but never speculate as to how 4-OI and NRF2 signaling could cause this to happen?

This is a deliberate choice of the authors not to speculate on the mode of action of 4-OI/Nrf2 on preventing IRF3 dimerization as it is some content of future work by the laboratory. Furthermore, we felt that these explanations would go beyond the scope of the main findings of this manuscript.

9. Lines 342-343. The statement on NRF2 deficiency in humans seems premature and speculative and likely should be deleted.

We acknowledge this position by the reviewer and have modified the text.

10. Fig 2a has a reduction of viral RNA to 6 log10 with 125 uM of 4-OI but the same dose in Fig 2b has a reduction to 3log10. What is the basis for the 1,000-fold discrepancy?

Each experiment has been performed at a different time point after infection with SARS-CoV-2. In (A) cells were harvested after 48h of infection while in (B) infected cells were collected at 24h post-infection. Figure legend has been edited accordingly.

Reviewer #2 (Remarks to the Author):

In this paper, Olganier et. al. show that activation of NRF2, an oxidative stress responder, can inhibit replication of both RNA and DNA viruses including SARS-CoV2. The authors test multiple viruses in human and non-human cell lines. They also show that siRNA-mediated knockdown of NRF2 results in increased viral replication while treatment of cells with NRF2 agonists 4-OI and DMF result in reduced viral replication. This antiviral activity is seen in the absence of STAT1 signaling suggesting that it is IFN-independent. However, the authors also present convincing data that NRF2 activation can suppress production of inflammatory cytokines including CXCL10. Given the strong correlation between CXCL10 and viral load in rhinovirus and SARSCoV2 infections (Cheermala, 2020 MedRxiv, Landry, 2018, JID), the data showing 4-OI mediated reduction of CXCL10 in patient blood samples is exciting. This is an important contribution to the field of NRF2 antiviral activity and potential SARS CoV2 antivirals.

I would recommend acceptance with minor revisions. I think the authors need to address some previous findings on NRF2 antiviral activity and restate some findings in the text. Specifically, the authors make a connection between NRF2-mediated suppression of IRF3 dimerization and reduction in inflammatory cytokine production and suppression of RIG-I signaling. However, this could also occur via other pathways including NFkB signaling which is not tested. MAVS is also shown to be suppressed by NRF2 in the absence of RIG-I activation which could also explain the suppression of RIG-I and STING signaling. In the absence of experiments targeting IRF3 activation (dimerization or phosphorylation) and showing subsequent reduction in inflammatory cytokine signaling, these overstatements should be modified (examples include lines 107-108, 294-295 etc.)

We were extremely pleased to read the comments made by Reviewer 2 on our manuscript who suggested an acceptance under minor revisions. Reviewer 2 raised a few minor concerns that we are addressing in the point-by-point response below.

Minor points:

1) Mihaylova et. al. 2018 showed that NRF2 antagonizes RIG-I activation in rhinovirus infection and that basal NRF2 activation varies in nasal epithelia vs. basal bronchial epithelial cells. Since these cells are also infected by SARS CoV2 discussion of these findings is highly relevant in this paper.

We acknowledge the point raised by Rev.2 which is now included in the discussion section of our revised manuscript

2) In Figure 1b, there is a set of genes activated in SARS-CoV2 biopsy and suppressed in all the cell lines – are RIG-I/ TLR signaling in this cluster? Also, the three dotted lines suggest three distinct clusters which are not highlighted in the text list of the genes.

This is a valid point raised by Reviewer 2. For further clarity, we have redesigned our Figure 1.b. Each cluster of genes is now appropriately referenced and the RIG-I/TLR signaling belongs to the pathways that are significantly enriched (red) in Covid-19 lung biopsies.

3) Diverging colour palette should have its center set at zero, please recolour figures 1b-c.

In accordance with reviewer 2, we have redesigned both Fig.1b and 1.d, so that the color palette is now centered in white towards zero

4) Please redraw Figure 1d so the gene names are readable. If the intent is to not have them be readable please remove the text.

The cloud map has also been redesigned so that gene names are now readable

5) Please show survival data for WT mice treated with 4-OI prior to HSV-2 infection (Figure S4).

As explained above and to address one of the points raised by reviewer 1, we decided to remove this supplementary *in vivo* figure from our revised manuscript as we were not able to get access to our animal facility to perform additional *in vivo* experiments during the Covid-19 lock down.

6) Please show individual data points in all the bar plots (example Figures 3h-j, 3l), for reference see Weissberger et. al., 2015, PloS Biology.

This is a valid point from Reviewer 2 and we have now changed the bar plots which now all show individual data points in the revised version of our document.

7) In Figure 3g, comparing NRF2 silenced+4-OI treated vs. 4-OI treated alone suggests that NRF2-independent pathways may also be responsible for 4-OI mediated antiviral activity. Is this difference significant?

In this experiment, the difference is indeed significant, for that reason we are not excluding the possibility that some of the antiviral effect mediated by the 4-OI is independent of Nrf2. We have softened our statement in the text to reflect this comment from Rev. 2.

Another point to consider here is also the partial silencing effect of Nrf2. Indeed, (1) 4-OI is by far the most potent Nrf2 inducer and (2) Nrf2 is a difficult gene/protein to target for transient silencing/gene editing. As seen, in this Fig. 3, the silencing of Nrf2 is only partial and cells still respond to 4-OI, as seen by HO-1 or Sqstm1 increase. This could also explain the quite modest phenotype that we report in Fig. 3g. To further validate this finding using another method, we performed a genetic activation of Nrf2 by silencing Keap1 in Calu-3 cells which were further infected with SARS-CoV-2. Here, genetic activation of Nrf2 led to a reduction in SARS infection/replication which is now reported in the revised Figure 2, panels s,t and u. Overall, we have adjusted our text to reflect the fact that at least some of the antiviral effects observed with the Nrf2 inducers is Nrf2-mediated.

8) In general, statistics are missing from multiple figures – for experiments that contain at least 3 replicates, statistics should be shown (see Figures 4a-d and others).

We agree with Reviewer 2 and statistics have now been added throughout the different figures of our manuscript

9) The legend for Figure 4d-f says that 'data are representative of one experiment performed in duplicates where means and s.e.m. are displayed.'. Does this mean that this experiment was only done once? If so, it needs to be repeated. If this is a representative experiment of multiple independent experiments then please restate the text.

In panels d-f, HAE cultures (n=4) were treated overnight with 4-OI at 125µM before SARS-CoV2 infection (MOI 0.1) for 24 and analyzed by qPCR. Data are representative of two independent experiments with two different donors. The figure legend has been edited accordingly.

10) Why are Figure 4g,h separate?

Cytokines measured by qPCR were sub-grouped based on their belonging to either the type I IFN pathway or the inflammatory pathway.

11) Please label cell type in western blot experiments (e.g. Figure 4n,o,q,t)

Cell types have been added to the revised figure 4

12) The figure axes and microscopy labels in Figure S2 are very hard to read, please resize.
This request from Rev.2 has been modified accordingly

Reviewers' Comments:

Reviewer #1:

Remarks to the Author:

The authors have responded thoughtfully to all of the critiques of this Reviewer and done their best to address the major concerns. As such, some of the conclusions about the direct role of NRF2 in mediating the effect of 4-OI and DMF have been tempered. The paper is now much stronger and provides a mechanistic framework for testing whether fumarate based therapies modulate SARS-CoV-2 infection and inflammation in vivo in animal models and possibly in humans.

Reviewer #1 (Remarks to the Author):

The authors have responded thoughtfully to all of the critiques of this Reviewer and done their best to address the major concerns. As such, some of the conclusions about the direct role of NRF2 in mediating the effect of 4-OI and DMF have been tempered. The paper is now much stronger and provides a mechanistic framework for testing whether fumarate based therapies modulate SARS-CoV-2 infection and inflammation in vivo in animal models and possibly in humans.

We would like to thank Reviewer 1 for his/her critical assessment of the revised version of our work and are delighted to read that our revisions have fulfilled his/her expectations.